# Robotic versus laparoscopic distal gastrectomy for resectable gastric cancer: a randomized phase 2 trial

Jun Lu[1,2,3,4,5], Bin-bin Xu[1,2,3,4,5], Hua-Long Zheng[1,2,3,4,5], Ping Li [1,2,3,4], Jian-wei Xie[1,2,3,4], Jia-bin Wang[1,2,3,4], Jian-xian Lin[1,2,3,4], Qi-yue Chen[1,2,3,4], Long-long Cao[1,2,3,4], Mi Lin[1,2,3,4], Ru-hong Tu[1,2,3,4], Ze-ning Huang[1,2,3,4], Ju-li Lin[1,2,3,4], Zi-hao Yao[1,2,3,4], Chao-Hui Zheng [1,2,3,4] & Chang-Ming Huang [1,2,3,4]

Robotic surgery may be an alternative to laparoscopic surgery for gastric cancer (GC). However, randomized controlled trials (RCTs) reporting the differences in survival between these two approaches are currently lacking. From September 2017 to January 2020, 300 patients with cT1-4a and N0/+ were enrolled and randomized to either the robotic (RDG) or laparoscopic distal gastrectomy (LDG) group (NCT03313700). The primary endpoint was 3-year disease-free survival (DFS); secondary endpoints reported here are the 3-year overall survival (OS) and recurrence patterns. The remaining secondary outcomes include intraoperative outcomes, postoperative recovery, quality of lymphadenectomy, and cost differences, which have previously been reported. There were 283 patients in the modified intention-to-treat analysis (RDG group: $n = 141$; LDG group: $n = 142$). The trial has met pre-specified endpoints. The 3-year DFS rates were 85.8% and 73.2% in the RDG and LDG groups, respectively ($p = 0.011$). Multivariable Cox regression model including age, tumor size, sex, ECOG PS, lymphovascular invasion, histology, pT stage, and pN stage showed that RDG was associated with better 3-year DFS (HR: 0.541; 95% CI: 0.314-0.932). The RDG also improved the 3-year cumulative recurrence rate (RDG vs. LDG: 12.1% vs. 21.1%; HR: 0.546, 95% CI: 0.302-0.990). Compared to LDG, RDG demonstrated non-inferiority in 3-year DFS rate.

Gastric cancer (GC) is among the most prevalent malignancies globally, ranking as the fifth most common cancer in terms of incidence and the fourth leading cause of cancer-related deaths[1]. Despite recent advancements in comprehensive treatment, complete tumor resection and lymphadenectomy remain the only effective approaches for the curative treatment of GC.

Over the past two decades, the surgical approach to GC has gradually shifted from laparotomy to minimally invasive procedures. Since Kitano's initial report on the application of laparoscopic gastrectomy for GC in 1994[2], laparoscopic surgery has been widely accepted by surgeons owing to its minimal invasiveness. Several multicenter trials have demonstrated that laparoscopic distal

[1]Department of Gastric Surgery, Fujian Medical University Union Hospital, Fuzhou, China. [2]Key Laboratory of Ministry of Education of Gastrointestinal Cancer, Fujian Medical University, Fuzhou, China. [3]Fujian Key Laboratory of Tumor Microbiology, Fujian Medical University, Fuzhou, China. [4]Fujian Province Minimally Invasive Medical Center, Fuzhou, China. [5]These authors contributed equally: Jun Lu, Bin-bin Xu, Hua-Long Zheng. ✉e-mail: wwkzch@163.com; hcmlr2002@163.com

**Fig. 1 | Trial profile.** Randomized controlled trial flowchart.

**Table 1 | Baseline Demographic and Postoperative Characteristics of Patients Who Underwent Robotic or Laparoscopic Surgery**

| Characteristic | RDG (n = 141) Mean ± SD/N (%) | LDG (n = 142) Mean ± SD/N (%) |
|---|---|---|
| Age, y | 59.4 ± 10.2 | 59.3 ± 11.3 |
| BMI, kg/m2 | 23.2 ± 3.0 | 22.7 ± 3.3 |
| Tumor size, mm | 35.1 ± 17.9 | 39.3 ± 19.6 |
| Sex | | |
| Male | 94 (66.7%) | 90 (63.4%) |
| Female | 47 (33.3%) | 52 (36.6%) |
| ECOG PS | | |
| 0 | 116 (82.3%) | 111 (78.2%) |
| 1 | 25 (17.7%) | 31 (21.8%) |
| Histology | | |
| Differentiated | 52 (36.9%) | 56 (39.4%) |
| Undifferentiated | 89 (63.1%) | 86 (60.6%) |
| Lymphovascular invasion | | |
| Negative | 81 (57.4%) | 76 (53.5%) |
| Positive | 60 (42.6%) | 66 (46.5%) |
| cT stage | | |
| cT1-T3 | 94 (66.7%) | 83 (58.5%) |
| cT4 | 47 (33.3%) | 59 (41.5%) |
| cN stage | | |
| cN0 | 63 (44.7%) | 58 (40.8%) |
| cN+ | 78 (55.3%) | 84 (59.2%) |
| pT stage | | |
| pT1 | 55 (39.0%) | 37 (26.1%) |
| pT2 | 23 (16.3%) | 19 (13.4%) |
| pT3 | 45 (31.9%) | 59 (41.5%) |
| pT4 | 18 (12.8%) | 27 (19.0%) |
| pN stage | | |
| pN0 | 54 (38.3%) | 56 (39.4%) |
| pN1 | 26 (18.4%) | 22 (15.5%) |
| pN2 | 28 (19.9%) | 27 (19.0%) |
| pN3a | 23 (16.3%) | 22 (15.5%) |
| pN3b | 10 (7.1%) | 15 (10.6%) |
| AJCC8th pTNM stage | | |
| I | 55 (39.0%) | 43 (30.3%) |
| II | 33 (23.4%) | 36 (25.4%) |
| III | 53 (37.6%) | 63 (44.4%) |

*RDG* robotic distal gastrectomy, *LDG* laparoscopic distal gastrectomy, *SD* Standard deviation, *BMI* Body mass index, *ECOG PS* Eastern Cooperative Oncology performance status, *AJCC* American Joint Committee on Cancer.

gastrectomy (LDG) for GC provides better short-term outcomes than laparotomy, without compromising survival[3–8]. However, laparoscopy has several technical limitations. For instance, it only provides a basically 2D surgical view, and the assessment of intraoperative positioning and spatial depth largely depends on the surgeon's experience[9]. There are also limitations related to restricted instrument movement, magnification of hand tremors, diminished tactile feedback, and nonergonomic surgical environment[10,11]. In contrast, the da Vinci robotic system offers several advantages, including operator-controlled 3D high-definition visualization, greater instrument maneuverability, tremor elimination, and improved ergonomics[10]. The complex anatomy and precise lymphadenectomy required for radical gastrectomy make these advantages particularly noteworthy.

Studies have confirmed that robotic gastrectomy (RG) offered less intraoperative blood loss, postoperative complications, and more retrieved lymph nodes (LN) than laparoscopic gastrectomy (LG)[12–16]. These findings were supported by two randomized controlled trials (RCTs)[16,17]. Nevertheless, only a limited number of retrospective studies reported differences in the long-term survival between the two approaches[18,19]. Therefore, using our expertise in robotic and laparoscopic gastrectomy for GC, we conducted this RCT to compare the effects of robotic distal gastrectomy (RDG) versus LDG. Safety outcomes revealed that RDG significantly reduced the overall postoperative complications compared with LDG while also increasing the harvested LNs and reducing the LN noncompliance rate[17].

Herein, we report the primary (i.e., 3-year disease-free survival) and secondary (i.e., 3-year overall survival and recurrence patterns) endpoints of this RCT to confirm the long-term oncological efficacy of RDG, which can provide high-level evidence to support the promotion of robotic surgery for GC.

## Results

### Study population
From July 2017 to January 2020, 300 patients were included and randomly assigned to two groups, with each group consisting of 150 patients. (Fig. 1). In the RDG group, seven patients who underwent total gastrectomy and two patients who withdrew their informed consent were excluded. In the LDG group, five patients who underwent total gastrectomy, two patients who had non-resectable tumors, and one patient with palliative surgery were excluded. Ultimately, there were 141 and 142 patients in the RDG and LDG groups, respectively.

There were 94 (66.7%) and 90 (63.4%) male patients in the RDG and LDG groups, respectively. The mean age was 59.4 years (SD, 10.2 years) in the RDG group and 59.3 years (SD, 11.3 years) in the LDG group. All these patients underwent R0 resection with D2

lymphadenectomy. No patients received postoperative radiotherapy. (Table 1).

The secondary outcomes included 3-year overall survival rate, 3-year recurrence pattern, overall postoperative morbidity rates, intraoperative morbidity rates, overall postoperative serious morbidity rates, number of retrieved lymph nodes, noncompliance rate of lymphadenectomy, time to first ambulation, time to first flatus, time to first liquid diet, time to first soft diet, duration of postoperative hospital stay, the variation of weight, the variation of cholesterol, the variation of album, the variation of white blood cell count, the variation of hemoglobin, hospitalization expenses and operation time. Here, the 3-year overall survival rate and recurrence pattern are reported. The remaining outcomes have been reported previously[17], apart from three (variation of weight, cholesterol, and album on

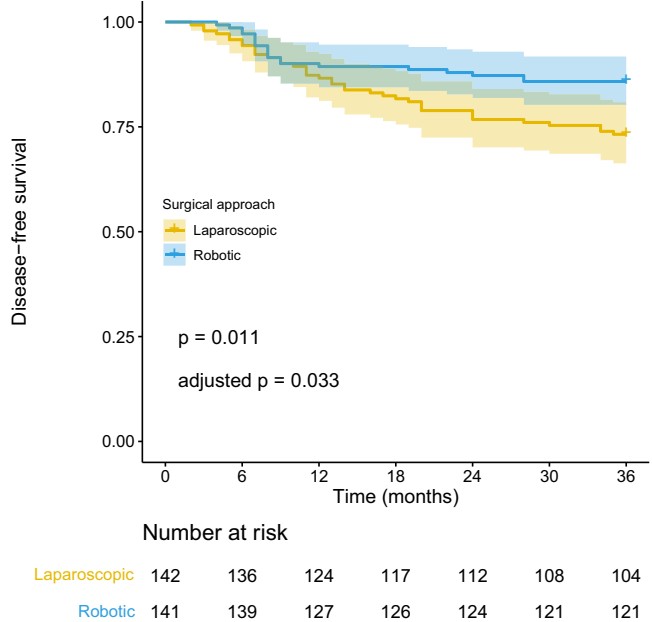

**Fig. 2 | Kaplan-Meier curves of disease-free survival for robotic and laparoscopic distal gastrectomies within 3 years after surgery.** The shadows on either side of the survival curves indicate 95% confidence intervals. p-values for all survival analyses have been calculated using the log-rank test. Adjusted p value was calculated using the Benjamini–Hochberg method. Source data are provided as a Source Data file.

postoperative 3, 6, 9, and 12 months) which are not reported as the information was only partially recorded.

**Primary outcome: 3-year disease-free survival**
Patients were followed up until death or for at least 36 months, with the last follow-up conducted on April 30, 2023. The median follow-up period was 48.0 months (95% CI: 45.7-50.3), with a total of 3 patients (1.1%) lost to follow-up (1 in the RDG group and 2 in the LDG group). The 3-year DFS rate in the RDG group was 85.8% (95% CI: 80.2–91.8%), whereas in the LDG group, it was 73.2% (95% CI: 66.3–80.9%). The absolute between-group difference (AD) was 12.6% (95% CI: 3.3–21.9%), which did not cross the prespecified non-inferiority margin of −16%, thus satisfying the primary hypothesis of non-inferiority of the RDG as compared with the LDG at 36-month follow-up. Furthermore, given that the two-sided 95% CI lies wholly to the right of zero, the RDG might be even superior to LDG, although there is a need for specific confirmatory superiority trials[20]. (Fig. 2). Stratified analysis based on pT and pN stage showed the following results: for patients with pT1 stage, the 3-year DFS rates were 96.4% (95% CI: 91.5–100%) vs. 97.3% (95% CI: 92.2–100%) (AD: −0.9%, 95% CI: −8.1% to 6.3%) for the RDG and LDG groups, respectively; for patients with pT2-4 stage, the rates were 79.1% (95% CI: 70.9–88.2%) vs. 64.8% (95% CI: 56.2–74.6%) (AD: 14.3%, 95% CI: 1.8% to 26.9%); for patients with PN0 stage, the rates were 100.0% vs. 92.9% (95% CI: 86.4–99.9%) (AD: 7.1%, 95% CI: 0.4% to 13.9%); and for patients with pN+ stage, the rates were 77.0% (95% CI: 68.7–86.4%) vs. 60.5% (95% CI: 51.0–71.7%) (AD: 16.5%, 95% CI: 2.9% to 30.1%) (Fig. 3A–D).

Baseline variables that were considered clinically relevant with outcomes were tested for proportional hazards assumption. All the variables satisfied the proportional hazards assumption (Supplementary Table 2). Multivariable Cox regression model including age, tumor size, sex, ECOG PS, lymphovascular invasion, histology, pT stage and pN stage showed that RDG remained an independent protective factor for the 3-year DFS (HR: 0.541; 95% CI: 0.314-0.932). (Table 2).

Figure 4 shows the effect size of RDG compared with LDG on the 3-year DFS in different subgroups. Overall, RDG seemed to show a trend of superiority over LDG in various subgroups.

Landmark analysis for the entire patient population revealed no significant difference in DFS between the two groups [RDG vs. LDG: 90.1% (95% CI: 85.3–95.1%) vs. 87.3% (95% CI: 82.0–93.0%), AD: 2.7%, 95% CI: −4.6-10.1%] at the 1-year follow-up. However, a difference was observed in the follow-up results between 1 to 3 years [RDG vs. LDG: 95.3% (95% CI: 91.7–99.0%) vs. 83.9% (95% CI: 77.6–90.6%), AD: 11.4%, 95% CI: 4.0–18.9%]. Similar results were observed in patients with pT2-4 and pN+ stages (Supplementary Fig. 1A–C).

**3-year overall survival**
The 3-year OS rate in the RDG group was 88.7% (95% CI: 83.6–94.0%), whereas that in the LDG group was 78.0% (95% CI: 71.4–85.2%) (AD: 10.7%, 95% CI: 2.0–19.3%) (Supplementary Fig. 2). For patients with pT1 stage, the 3-year OS rates were 98.2% (95% CI: 94.7–100.0%) vs. 97.3% (95% CI: 92.0–100.0%) (AD: 1.0%, 95% CI: −5.5-7.4%) for the RDG and LDG groups, respectively; for patients with pT2-4 stage, the rates were 82.6% (95% CI: 74.9–91.0%) vs. 71.3% (95% CI: 63.1–80.5%) (AD: 11.2%, 95% CI: −0.6–23.1%); for patients with PN0 stage, the rates were 100.0% vs. 92.9% (95% CI: 86.2–99.9%) (AD: 7.2%, 95% CI: 0.4–14.0%); for patients with pN+ stage, the rates were 81.6% (95% CI: 73.9–90.2%) vs. 68.6% (95% CI: 59.3–79.0%) (AD: 13.2%, 95% CI: 0.4–25.9%) (Supplementary Fig. 3A–D). The multivariable analysis showed that the HR for overall mortality in the RDG group was 0.542 (95% CI: 0.296–0.994) compared with the LDG group (Supplementary Table 3). There was a difference in the cumulative incidence of GC-specific death between the two groups (RDG vs. LDG: HR: 0.485, 95% CI: 0.249-0.944) (Table 3).

**Recurrence patterns**
During the 3-year follow-up period, there were 17 cases (cumulative incidence rate of 12.1%) of recurrence in the RDG group and 30 cases (cumulative incidence rate of 21.1%) in the LDG group (Table 3). When considering death as a competing risk factor, a significantly lower cumulative incidence of recurrence was observed in the RDG groups (HR: 0.546, 95% CI: 0.302–0.990). Stratified analysis of recurrence patterns indicated a difference in the cumulative local recurrence rate between the two groups (RDG vs. LDG: 2.1% vs. 7.7%; HR: 0.266, 95% CI: 0.075–0.950). No significant difference was observed in the cumulative rates of peritoneal recurrence and liver metastasis between the two groups (Supplementary Fig. 4).

**Number of LNs Examined and Rate of LN Noncompliance.** Generally, the total number of lymph nodes (LNs) dissected was comparable between the two groups [RDG vs. LDG: 40.9 ± 11.2 vs. 39.9 ± 12.2, absolute difference: 1.0, 95% CI: (−1.7–3.8)]. The proportion of patients with over 30 lymph nodes dissected was slightly higher in the RDG group (RDG vs. LDG: 85.8% vs. 78.2%, absolute difference: 7.6%, 95% CI: (−1.3%, 16.6%)]. Further stratified analysis by dividing LNs into perigastric regions (No. 1, 3, 4, 5, 6) and extra gastric regions (No. 7, 8a, 9, 11p, 12a) revealed that the number of LNs dissected in extraperigastric regions was significantly higher in the RDG group (RDG vs. LDG: 17.6 ± 5.8 vs. 15.8 ± 6.6, absolute difference: 1.8, 95% CI: 0.3–3.2). The comparison of LN noncompliance between the 2 groups indicated that the LN noncompliance rate of the RDG group was significantly lower than that of the LDG group [RDG vs. LDG: 24.8% vs. 40.1%, absolute difference: −15.3%, 95% CI: (−26.1%, −4.6%)]. (Supplementary Table 4).

**Adjuvant chemotherapy characteristics.** Supplementary Table 5 presented information regarding postoperative adjuvant chemotherapy for both groups. Seventy-three patients (91.25%) in the LDG group received docetaxel based chemotherapy regimen, compared to 61 patients (89.7%) in the RDG group (absolute difference: 1.5%, 95% CI:

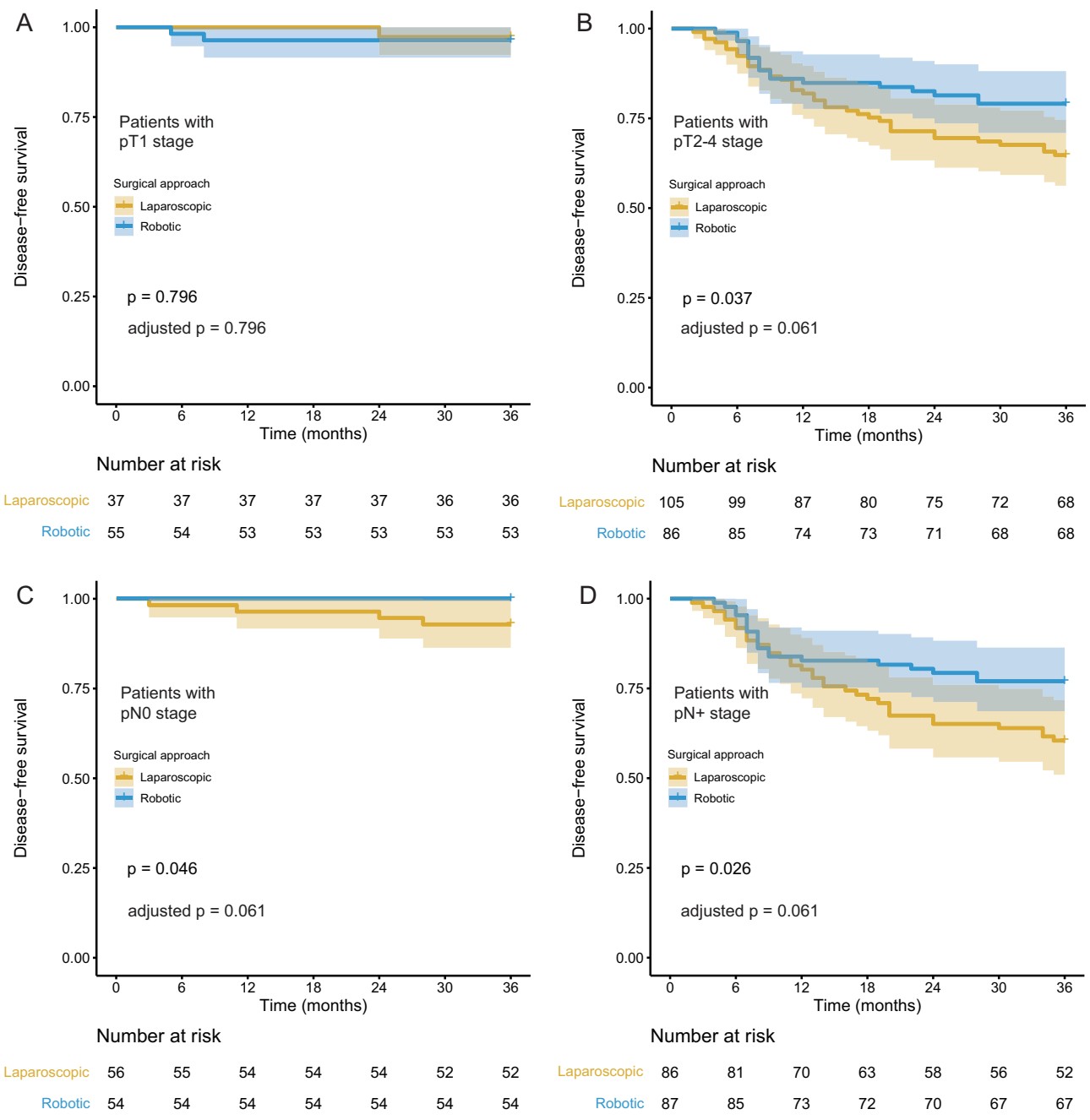

**Fig. 3 | Kaplan-Meier curves of disease-free survival for robotic and laparoscopic distal gastrectomies within 3 years after surgery by different pathologic T stage and N stage. A** patients with pT1 stage; **B** patients with pT2-4 stage; **C** patients with pN0 stage; **D** patients with pN+ stage. *p*-values for all survival analyses have been calculated using the log-rank test. The shadows on either side of the survival curves indicate 95% confidence intervals. Adjusted p value was calculated using Benjamini–Hochberg method. Source data are provided as a Source Data file.

−8.0−11.1%). Aside from the median duration in days between surgery and chemotherapy [RDG vs. LDG: 28 vs. 32, AD: −4, 95% CI: (−7, −1)], there was no statistical differences between the two groups in terms of chemotherapy cycles and toxicity.

## Discussion

This study is the RCT to compare the long-term oncological outcomes of RDG and LDG in patients with GC. RDG demonstrated noninferiority in 3-year DFS rate. Significant differences were observed between the two groups in terms of cumulative recurrence rates. Furthermore, the advantages of RDG technology seem more pronounced after one year of follow-up. These findings provide high-level

evidence to support the further promotion of robotic surgery for patients with GC.

Despite the proven safety and feasibility of laparoscopic surgery for GC in recent years, it has some limitations. Laparoscopic instruments have limited mobility, restricted visual field, and only provide a basically 2D view, which lacks depth perception. However, two large-scale meta-analyses involving robotic metabolic and bariatric surgery indicated that, compared to laparoscopic surgery, robotic metabolic and bariatric surgery (sleeve gastrectomy, Roux-en-Y gastric bypass) extended the operation time without significantly improving postoperative recovery[21,22]. From a technical perspective, the inherent advantages of robotic systems, such as higher magnification, 3D

imaging, and more precise manipulation of organs, blood vessels, and nerves, are more appealing for complex and challenging radical gastrectomies. A large-scale meta-analysis involving more than 17,000 gastric cancer patients also found that the operation time for robotic gastrectomy was significantly longer than laparoscopic surgery, but

**Table 2 | Multivariable Cox Regression Analyses of Risk Factors for Disease-free Survival**

| | Multivariate Analysis | | |
|---|---|---|---|
| | HR | 95%CI | p value |
| Age, ≥65 | – | – | 0.381 |
| Tumor size, mm | 1.014 | 1.000-1.028 | 0.048 |
| Female | – | – | 0.934 |
| ECOG PS, score 1 | – | – | 0.157 |
| Surgical approach, robot | 0.541 | 0.314-0.932 | 0.027 |
| Lymphovascular invasion, yes | – | – | 0.226 |
| Histology, undifferentiated | – | – | 0.354 |
| pT stage | | | |
| T1 | 1.000 | | |
| T2-4 | 3.524 | 1.035-11.997 | 0.044 |
| pN stage | | | |
| N0 | 1.000 | | |
| N+ | 5.576 | 1.940-16.023 | 0.001 |
| Adjuvant chemotherapy, yes | – | – | 0.502 |

*ECOG PS* Eastern Cooperative Oncology performance status.

the intraoperative blood loss and postoperative complication rates were lower in the robotic group[23]. An RCT conducted at two centers in Japan demonstrated that RG reduced the occurrence of postoperative complications and decreased the dosage of postoperative analgesics compared with LG[16]. Our previous reports also indicated that patients in the RDG group had significantly lower rates of postoperative complications and faster postoperative recovery[17]. Nevertheless, further investigations on the long-term outcomes of RG are still required.

A propensity score-matched study from one single center in Korea, involving 2084 patients, showed a comparable 5-year OS (RG vs. LG: 93.2% vs. 94.2%, $p = 0.628$) and 5-year recurrence-free survival (RFS) (RG vs. LG: 95.3% vs. 96.3%, $p = 0.509$) between the two groups[19]. In that study, a higher proportion (approx. 80%) of early-stage patients (pT1) were included. In a retrospective study from seven centers in China, using propensity score matching, the robotic group had slightly higher long-term survival than the laparoscopic group with no statistical difference. However, the median follow-up time in that study was less than 3 years[18]. For trials from Japan, a multi-institutional study showed better 3-year OS for patients with $p$ stage IA GC[24]. However, another study from Japan showed that when compared with LG, RG significantly improved the 5-year OS and 5-year RFS for patients with GC, especially for patients with advanced stage[25]. Therefore, the aforementioned results lack confirmation from prospective randomized controlled trials.

Based on our expertise in LG and RG for GC treatment, we conducted this RCT, in which patients were followed up for 3 years. The 3-year DFS in the RDG group was significantly better than that in the LDG group. Moreover, in multivariable analysis adjusted for various variables, RDG remained an independent protective factor for the 3-year DFS. Further analysis of the recurrence patterns revealed that

| Subgroup | Robotic | Laparoscopic | HR (95%CI) | Favors RDG | Favors LDG | Adjusted P value |
|---|---|---|---|---|---|---|
| | No. of events / total No. | | | | | |
| Age, y | | | | | | |
| <65 | 15/98 | 21/80 | 0.560 (0.289-1.087) | | | 0.134 |
| ≥65 | 5/43 | 17/62 | 0.393 (0.145-1.066) | | | 0.125 |
| Sex | | | | | | |
| Male | 13/94 | 25/90 | 0.471 (0.241-0.921) | | | 0.095 |
| Female | 7/47 | 13/52 | 0.573 (0.229-1.437) | | | 0.307 |
| ECOG PS | | | | | | |
| 0 | 19/116 | 29/111 | 0.603 (0.338-1.076) | | | 0.134 |
| 1 | 1/25 | 9/31 | 0.125 (0.016-0.986) | | | 0.117 |
| Lymphovascular invasion | | | | | | |
| Negative | 6/81 | 9/76 | 0.625 (0.222-1.755) | | | 0.424 |
| Positive | 14/60 | 29/66 | 0.473 (0.250-0.896) | | | 0.094 |
| Histology | | | | | | |
| Differentiated | 7/52 | 9/56 | 0.839 (0.312-2.254) | | | 0.728 |
| Undifferentiated | 13/89 | 29/86 | 0.395 (0.205-0.761) | | | 0.085 |
| Tumor size, cm | | | | | | |
| <5 | 11/112 | 19/98 | 0.488 (0.232-1.026) | | | 0.125 |
| ≥5 | 9/29 | 19/44 | 0.697 (0.315-1.543) | | | 0.424 |
| Postoperative complication | | | | | | |
| No | 17/128 | 30/117 | 0.490 (0.270-0.888) | | | 0.094 |
| Yes | 3/13 | 8/25 | 0.720 (0.191-2.717) | | | 0.667 |
| Adjuvant chemotherapy | | | | | | |
| No | 5/59 | 9/52 | 0.477 (0.160-1.424) | | | 0.262 |
| Yes | 15/82 | 29/90 | 0.531 (0.285-0.991) | | | 0.117 |
| Overall | 20/141 | 38/142 | 0.504 (0.293-0.866) | | | 0.094 |

0.01    0.11    1.00    9.00
HR (95%CI)

**Fig. 4 | Subgroup analysis of disease-free survival including age, sex, ECOG PS, lymphovascular invasion, histology, tumor size, postoperative complication, and adjuvant chemotherapy (RDG [*n* = 141] vs. LDG [*n* = 142]).** Forest plots show the hazard ratio (HR) as centers, and the upper and lower hinges represent the corresponding 95% confidence intervals (CIs). A Cox proportional hazard model without stratification factors was used to calculate HRs for group comparisons. *P* values were two-sided at the 5% significance level and adjusted by the Benjamini–Hochberg method. Source data are provided as a Source Data file.

**Table 3 | Frequencies of Causes of First Recurrence and Death Within 3 Years After Surgery in Patients Who Underwent Robotic or Laparoscopic Surgery**

| Events | No. (%) | | Risk Difference[a] | Hazard Ratio (95% CI)[b] | p value[c] | Adjusted p value[d] |
|---|---|---|---|---|---|---|
| | RDG (n = 141) | LDG (n = 142) | | | | |
| Any recurrence[e] | 17 (12.1%) | 30 (21.1%) | 0.092 | 0.546 (0.302-0.990) | 0.046 | 0.046 |
| Local | 3 (2.1%) | 11 (7.7%) | 0.056 | 0.266 (0.075-0.950) | 0.041 | 0.144 |
| Peritoneum | 5 (3.5%) | 8 (5.6%) | 0.021 | 0.627 (0.206-1.911) | 0.412 | 0.535 |
| Liver | 4 (2.8%) | 6 (4.2%) | 0.014 | 0.670 (0.190-2.379) | 0.535 | 0.535 |
| Multiple sites[f] | 6 (4.3%) | 9 (6.3%) | 0.021 | 0.665 (0.238-1.862) | 0.438 | 0.535 |
| Other or uncertain sites[g] | 9 (6.4%) | 17 (12.0%) | 0.056 | 0.518 (0.232-1.159) | 0.109 | 0.254 |
| Cause of death[h] | 16 (11.3%) | 31 (21.8%) | 0.107 | 0.498 (0.273-0.911) | 0.024 | NA[i] |
| Gastric cancer | 13 (81.3%) | 26 (83.9%) | 0.092 | 0.485 (0.249-0.944) | 0.033 | 0.144 |
| Other causes[j] | 3 (18.8%) | 5 (16.1%) | 0.014 | 0.594 (0.143-2.472) | 0.474 | 0.535 |

[a]Except for all-cause death, the risk difference was calculated by subtracting the cumulative incidence in the first 3 years of the robotic group from that of the laparoscopic group, in presence of competing events; for all-cause death, the risk difference was calculated by subtracting the 3-year overall survival rate of the robotic group from that of the laparoscopic group.

[b]Except for all-cause death, competing-risks survival regression was used to derive the hazard ratio, 95% CI, and P value. For total recurrence, all-cause death was the competing event; for the specific types of recurrence, other types of recurrence and death were the competing events; for gastric cancer cause of death, other causes of death were the competing events, and vice versa. Univariate Cox regression was used for all-cause death.

[c]P values for the hazard ratios.

[d]P values adjusted by Benjamini–Hochberg method.

[e]Refers only to first-time recurrence, even though patients can have recurrence at multiple times.

[f]Includes patients who have recurrence simultaneously in 2 or more metastatic sites, including peritoneum, liver, lung, bone, brain, distant lymph node, or other hematogenous metastatic sites

[g]Includes hematogenous recurrence at sites other than liver (ie, lung, bone, brain), recurrence at distant lymph node, and recurrence at uncertain sites.

[h]Post hoc exploratory outcomes.

[i]Univariate Cox regression was used for all-cause death.

[j]Includes other cancers, diseases other than cancer, unintentional injuries, and unknown causes.

RDG robotic distal gastrectomy, LDG laparoscopic distal gastrectomy, CI confidence interval, NA not applicable.

RDG significantly reduced the cumulative incidence of local recurrence for patients after surgery. These results may be attributed to the following reasons:

First, existing studies have shown that the likelihood of cancer cell dissemination may increase as surgical bleeding increases[26], and our previous reports found that RDG significantly reduced the intraoperative bleeding[17], which is consistent with most previous studies. Second, the occurrence of postoperative complications has been found to be associated with poor prognosis for GC[27,28]. Previous reports on the short-term outcomes of this study indicated that the overall postoperative complication rate was significantly lower in the RDG group than in the LDG group (RDG vs. LDG: 9.2% vs. 17.6%, absolute difference: −8.4%, 95% CI: −16.3% to −0.5%). Based on the ClavienDindo classification system, a significant difference was observed in the grade II complications between the 2 groups (RDG vs. LDG: 7.8% vs. 15.5%, absolute difference: −7.7%, 95% CI: −15.1−−0.3%)[17]. The mechanism affecting the prognosis may be related to more pronounced postoperative systemic inflammation and severe immunosuppression. Infectious complications and sepsis can enhance the cascade of proinflammatory cytokines, including tumor necrosis factor-alpha and interleukins 1, 6, and 8. These immunomodulators have the ability to impact the function and regulation of natural killer cells, cytotoxic T lymphocytes, and antigen-presenting cells[29–31]. Furthermore, postoperative complications result in prolonged immune suppression, allowing residual tumor cells to proliferate and survive longer in the host and promote the occurrence of micrometastases[27], thereby impacting the long-term survival of patients.

Furthermore, previous reports have indicated that the RDG group retrieved more LNs from extraperigastric stations and a lower noncompliance rate of LN dissection than the LDG group[17]. Currently, studies have shown that the noncompliance rate of LN dissection significantly affects the prognosis of patients with GC[32,33]. Given the inherent advantages of the robotic surgical system, such as 3D high-definition imaging, elimination of hand tremors, and articulating instruments, it can enable the thorough removal of deeply located extraperigastric lymph nodes. Therefore, RDG may reduce the incidence of local recurrence and provide more accurate staging information for clinical decision-makers. Previous studies indicated that the incidence of local recurrence in patients undergoing D2 curative gastrectomy varies from 3% to 20%[3,18,34–41]. In studies calculating postoperative recurrence rates, there are two methods of enumeration. One approach involves repeated counting. As some patients have recurrences in different locations at the time of initial recurrence detection (i.e., multiple sites), these are counted repeatedly to provide a more comprehensive depiction of recurrence types[34–38]. This is the method of demonstrating recurrence patterns that we used in the present study, as shown in the Venn diagram (Supplementary Fig. 4). Another instance involves counting patients with simultaneous recurrences in different locations as a singular type of recurrence, without repeating the count with other types[3,18,39–41]. Different methods of enumeration lead to variations in recurrence patterns, and in several studies including CLASS 01, local recurrence remains the most common type of recurrence[3,18,41]. When repeated counting is not taken into account, the local recurrence rate in the present study is 4.9% (7/142) (Supplementary Fig. 4), which is similar to previous studies[3,18,39–41].

Numerous studies have demonstrated that adjuvant chemotherapy (ACT) could significantly prolong the survival of patients with advanced GC[42,43]. While the optimal timing of ACT after radical gastrectomy remains a subject of debate, many studies have suggested that initiating ACT late is associated with worse survival for patients with GC[44–46]. In our study, although the rates of ACT and the number of completed cycles were comparable between the two groups, the interval between operation and ACT was significantly shorter in the RDG group[17]. This may also be one reason for the significantly better prognosis in the RDG group. Furthermore, it was surprising that the 3-year DFS curves of the two groups showed a noticeable crossover within the first 12 months. Subsequently, we performed landmark analyses with 1 year as a cutoff and found significant differences in DFS between the two groups from 1 to 3 years; similar results were observed in patients with pT2-4 and pN+. However, the statistical power may be insufficient to conduct further subgroup analyses owing to the small sample size. Further exploration of these findings in larger RCTs is necessary to elucidate the underlying reasons.

Our findings suggested that RDG could be even more efficacious than LDG with regard to the 3-year DFS and local recurrence rate. However, the past evidence suggested us to design this study as a non-inferiority trial. Several RCTs with a non-inferiority design also revealed a similar situation[47–49]. As a consequence, the superiority of the RDG should be verified with a specifically designed superiority trial and should be thus an object of future investigations. A post hoc calculation was performed based on our findings and, assuming a superiority margin of 12.6%, with a 3-year DFS of 73.2% in the control group and an expected rate in the experimental group of at least 85.6%, 161 per group would be needed, with a 90% power and alpha at 0.025. Given the proximity to the calculated sample size, therefore, the results of this study, which portends towards a potential of superiority of RDG, are of great importance in clinical practice and suggest that we design a superiority trial based on the evidence that emerged from the present study.

Several limitations should be acknowledged in the present study. First, all operations were performed by experienced surgeons at a high-volume hospital in the Eastern region. Hence, the general applicability of our findings may be constrained to surgeons with varying levels of expertise. Second, this RCT only enrolled patients undergoing distal gastrectomy, which may restrict the applicability of the results, particularly in Western countries where tumors in the upper stomach are more common[1,50]. Lastly, the economic and societal benefits of RG remain uncertain owing to the higher cost of robotic surgery; hence, a cost-effectiveness analysis is required.

In summary, as the RCT comparing robotic and laparoscopic gastrectomy for GC with long-term oncological outcomes, our study demonstrated non-inferiority of RDG versus LDG in terms of the 3-year DFS. This study provides evidence to support the utilization of RDG, performed by experienced surgeons, as a viable therapeutic option for patients with tumors located in the lower stomach, in addition to conventional laparoscopic surgery. Further, multicenter RCTs with a superiority design are required to validate these findings.

## Methods

### Study design

The current study was conducted from September 2017 to January 2020 at a tertiary referral teaching hospital performing over 800 gastrectomies each for GC in China[17]. The study was performed in accordance with the Declaration of Helsinki and Good Clinical Practice guidelines. The Ethics Committee of Fujian Medical University Union Hospital approved this RCT on September 22, 2017 (IRB number: 2017YF015-02). All of the patients were recruited after the RCT was approved. Prior to registration for Clinical Trial, from September 25 to October 10, a total of 13 patients were enrolled, including 6 patients in the LDG group and 7 patients in the RDG group. The last patient was enrolled on January 13, 2020. This study is a phase II, noninferiority, open-label, randomized clinical trial (ClinicalTrials.gov, NCT03313700). The inclusion criteria for this study are as follows: (1) Age from 18 to 75 years (not including 18 and 75 years old); (2) Primary gastric adenocarcinoma confirmed pathologically by endoscopic biopsy; (3) Clinical stage tumor T1-4a (cT1-4a), N0/+, M0 at preoperative evaluation according to the American Joint Committee on Cancer (AJCC) Cancer Staging Manual Eighth Edition; (4) Expected to undergo distal gastrectomy and D1+/D2 lymph node dissection to obtain R0 surgical results; (5) Performance status of 0 or 1 on the ECOG (Eastern Cooperative Oncology Group) scale; (6) ASA class I to III; (7) Written informed consent. The exclusion criteria for this study are as follows: (1) Women during pregnancy or breast-feeding; (2) Severe mental disorder; (3) History of previous upper abdominal surgery (except for laparoscopic cholecystectomy); (4) History of previous gastric surgery (including ESD/EMR for gastric cancer); (5) Multiple primary gastric cancer; (6) Enlarged or bulky regional lymph node diameter over 3 cm by

preoperative imaging; (7) History of other malignant disease within past five years; (8) History of previous neoadjuvant chemotherapy or radiotherapy; (9) History of unstable angina or myocardial infarction within the past six months; (10) History of cerebrovascular accident within past six months; (11) History of continuous systematic administration of corticosteroids within one month; (12) Requirement of simultaneous surgery for another disease; (13) Emergency surgery due to complications (bleeding, obstruction or perforation) caused by gastric cancer; (14) Forced expiratory volume in 1 second (FEV1) < 50% of the predicted values. Patients were provided with detailed information regarding potential risks associated with their participation and the additional costs of RG. Participants had the right to withdraw their consent at any point during the study without consequences to their medical care. Pathological staging was performed based on the 8th American Joint Committee on Cancer (AJCC) staging system. Details regarding the inclusion and exclusion criteria are available in the trial protocol provided in Supplementary Note 2.

### Surgical procedures and quality control

The Da Vinci robotic system (Intuitive Surgical, Inc., Sunnyvale, CA) was used by the same group of surgeons (CM.H and CH.Z) with experience in more than 300 LGs and 50 RGs for GC. D2 lymphadenectomy including No. 1, 3, 4sb, 4d, 5, 6, 7, 8a, 9, 11p, and 12a was performed according to the 4th edition of the Japanese Gastric Cancer Treatment Guidelines[51]. Another group of surgeons reviewed the unedited videos of participants' lymphadenectomies once a week using a sample survey (Supplementary Table 1) for standardization and quality control[17]. All the D2 gastrectomies were determined acceptable. Palliative surgery in this study is defined as non-curative gastrectomy or gastrojejunostomy performed to alleviate severe complications (such as bleeding or obstruction) caused by the tumor in patients with advanced or metastatic gastric cancer.

**Adjuvant chemotherapy.** We routinely recommended chemotherapeutic regimens based on 5-fluorouracil (5-FU) in combination with either platinum-based drugs or docetaxel for patients with pathological stage II or stage III.

### Outcome measurements

The primary endpoint of this trial is the 3-year disease-free survival (DFS), whereas the secondary endpoints include the 3-year overall survival (OS) and recurrence patterns.

DFS was defined as the time from surgery to recurrence or death from any cause. OS was defined as the time from surgery to death from any cause or the last follow-up. Recurrence was determined based on medical history and physical examination in combination with imaging evaluation, cytology, or tissue biopsy.

We divided the lymph node (LN) into 2 regions: perigastric regions (stations 1–6) and extraperigastric regions (stations 7–9, 11p, and 12a)[17]. Lymph node dissection noncompliance was defined as the absence of LNs from more than one LN station that should have been excised[17].

The secondary outcomes included 3-year overall survival rate, 3-year recurrence pattern, overall postoperative morbidity rates, intraoperative morbidity rates, overall postoperative serious morbidity rates, number of retrieved lymph nodes, noncompliance rate of lymphadenectomy, time to first ambulation, time to first flatus, time to first liquid diet, time to first soft diet, duration of postoperative hospital stay, the variation of weight, the variation of cholesterol, the variation of album, the variation of white blood cell count, the variation of hemoglobin, hospitalization expenses and operation time. Here, the 3-year overall survival rate and recurrence pattern are reported. The remaining outcomes have been reported previously[17], apart from three (variation of weight, cholesterol, and album on

postoperative 3, 6, 9, and 12 months) which are not reported as the information was only partially recorded.

## Follow-up

A minimum follow-up period of 36 months was mandated for each patient following gastrectomy. The patients underwent regular follow-up visits at specified intervals, including every 3 months during the first 2 years and every 6 months thereafter. The follow-up assessments encompassed various components including physical examinations, laboratory tests (peripheral blood routine assessment, blood biochemistry, serum tumor markers including CA19-9, CA72-4, and CEA level), chest radiography, abdominal computed tomographic scans, and annual endoscopic examinations. Should a recurrence be suspected during the aforementioned examinations, positron emission tomography/computed tomography scans were conducted to further investigate the condition.

## Randomization and masking

Eligible patients were randomly allocated in a 1:1 ratio to either LDG or RDG group. The data manager (RH.T.), who was not involved in the eligibility assessment and recruitment of patients, conducted the randomization using a randomly ordered treatment identifier list generated by a permuted block design with a block size of six using SAS software (version 9.2, SAS Institute Inc.). This implies that every six patients form a randomized block, with the treatment order within each block being random, and both treatment LAG and treatment RDG accounting for half of each block. Subsequently, these blocks are concatenated to form a list, and the treatment assignment for each patient is done sequentially according to this list. Because of the impracticality of implementing masking, the study was not blinded after randomization, and the patients were informed about the surgical approach. However, the trial coordinators (M.L. and J.B.W.) and pathologists were unaware of the assigned treatment.

## Adjustment for multiple comparisons

The present study aimed to compare the long-term outcomes of robotic distal gastrectomy (RDG) and laparoscopic distal gastrectomy (LDG). The prespecified endpoints in the present study included 3-year DFS, 3-year OS, and recurrence patterns. Therefore, the three p values were adjusted by the Benjamini-Hochberg adjustment method[52,53]. In addition, we also adjusted the p values in the subgroup analysis of the 3 endpoints independently.

## Statistics & reproductbility

There are no deviations in the analysis plan compared with the preregistered protocol. The modified intention-to-treat (MITT) analysis set population was used for all analyses in this study. The MITT group excluded patients who met exclusion criteria post-randomization. These criteria, as defined in the study protocol (Supplementary Note 2), included intraoperatively or post-operatively confirmed T4b or M1, unresectable tumors, total gastrectomy, and withdrawal of informed consent. The calculation of sample size was implemented using nQuery Advisor 7.0 (Cork, Ireland), which was described previously[17]. The 3-year DFS and OS rates were determined using the Kaplan-Meier method, and the significance was assessed using the log-rank test. Subgroup analyses were conducted for DFS and OS stratified by pathologic T (pT) stage (ie, pT1, pT2-4) and the LN status (ie, pN0, pN+). Baseline variables that were considered clinically relevant with outcomes were tested of proportional hazards assumption and then entered into the multivariate Cox proportional hazards model (forward stepwise). All-cause mortality was considered a competing event for recurrence and cumulative incidence was calculated. Furthermore, landmark analyses were conducted to evaluate the outcomes at 1 year and subsequent follow-up[54]. All analyses were performed with SPSS statistical software (version 21.0; SPSS Inc.) and R. All tests were two-sided, and $p < 0.05$ was considered statistically significant.

## Reporting summary

Further information on research design is available in the Nature Portfolio Reporting Summary linked to this article.

## Data availability

The data supporting the findings in this study are available under controlled access due to data privacy laws related to patient consent for data sharing and the data should be used for research purposes only. All the original clinical data will be made available on request from the corresponding author (Huang CM) at any time in a de-identified manner. Request for data sharing will be handled in line with the data access and sharing policy of Fujian Medical University Union Hospital, which can be found in Supplementary Note 1. The original study protocol is available as Supplementary Note 2 in the Supplementary information file. The remaining data are available within the Article, Supplementary Information, or Source Data file. Source data are provided in this paper.

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

## Acknowledgements

We thank those who have devoted a lot to this study, including nurses, pathologists, and doctors. Thanks to statistician Liu Fengqiong, from Fujian Medical University, for her guidance in statistics. This study was supported by the Fujian Province Medical "Creating high-level hospitals, high-level medical centers, and key specialty projects ([2021] No.76, H.C.M.), Yunnan Provincial Science and Technology Department (No. 202105AF150040, H.C.M.) and Fujian Research and Training Grants for Young and Middle-aged Leaders in Healthcare for Jun Lu. (No. [2023] 26, L.J.) The funding source had no role in the design and conduct of the study; collection, management, analysis, and interpretation of the data; preparation, review, or approval of the manuscript; and decision to submit the manuscript for publication.

## Author contributions

H.C.M. had full access to all the data in the study and took responsibility for the integrity of the data and the accuracy of the data analysis. H.C.M. and L.J. obtained funding, conceived of and designed the study, and supervised the whole study. P.L., J.-X., Q.-C., L.-l.C., M.L., R.-T., Z.-H.J., X.B.B., and H.C.M. drafted the manuscript. H.C.M., L.J., X.B.B., Z.H.L., and Z.C.H. critically revised the manuscript. All authors reviewed the manuscript and agreed to submit it for publication.

## Competing interests

The authors declare no competing interests.
