## [Peer Review File · Nature Communications]

Reviewers' Comments:

Reviewer #1:

Remarks to the Author:

The authors report the results of their prospective, randomized, clinical trial evaluating robotic vs laparoscopic gastrectomy for gastric cancer. They are commended for such a large population of a disease site that is difficult to study due to low prevalence in Western countries. While their findings are intriguing and consistent with prior reports favoring robotic surgery for lower surgical morbidity, there are several limitations of the manuscript that render it not acceptable to publication.

1. There is no information regarding the type of chemotherapy patients received postoperatively, how it was tolerated, toxicity, etc. Was chemotherapy prescribed per protocol? Type and cycles of chemotherapy may significantly influence oncologic outcomes, independent of surgical resection modality.
2. There are no data on how many patients underwent D1 vs D2 surgery. There are no information regarding number of lymph nodes that were dissected to evaluate if the oncologic surgery was adequate. There are no information on if postoperative radiation was used in any patients. All of these factors could significantly influence oncologic endpoints.
3. It is not clear to this reviewer why the authors list a meta-analysis, not written at their institution, to report their surgical quality (page 6, reference 9).
4. There are no data regarding follow-up compliance.

Reviewer #2:

Remarks to the Author:

Referee's report on Nature Communications manuscript NCOMMS-23-51139, "Effect of Robotic versus Laparoscopic Distal Gastrectomy on 3-Year Disease-Free Survival among Patients with Resectable Gastric Cancer: Outcomes from a Randomized Controlled Trial" by Lu et al.

Comments for the Authors

1. There are several published large scale retrospective meta-analyses comparing robotic surgery (RS) to laparoscopic gastric surgery that the introduction does not cite. I have listed 3 papers below. While the types of surgeries and goals of these meta-analyses differ, and these were large meta-analyses rather than a single prospective randomized trial, the results of these studies are highly relevant and should be contrasted with those reported in the present paper. A general result seems to be that robotic surgery takes longer, so a question is whether the trial reported in the present paper is confirming this, or if new or different results are being reported.

Guerrini et al. (*International Journal of Surgery*, 2020, 82:210-228) considered the comparison for gastric cancer in terms of intra-operative, post-operative, and oncological outcomes.

Leang et al. (*Surgery for Obesity and Related Diseases*, 2024, 20:62-71) considered bariatric surgery in terms of multiple adverse perioperative outcomes.

Vosberg et al. (*Obesity Surgery* (2022) 32:2341-2348) considered metabolic and bariatric surgery.

2. The modified ITT sample sizes of 141 and 142 strongly suggest that the randomization was restricted to achieve balance. Was this the case?

3. No goodness-of-fit analyses are reported for the Cox model. A potential problem is that, if the proportional hazards assumption was not met by the trial's data, then there would be no single hazard ratio (HR) parameter, and it would become necessary to identify a different regression model that provides an acceptable fit to the data, and report its results. In this case, the currently reported HR values would need to be replaced by appropriate parameter estimates.

4. More generally, the main reason for fitting a regression model in this sort of setting is to assess possible effects of patient baseline covariates on DFS time, in addition to the between-treatment effect. However, no covariate effects are even mentioned in the abstract.

4. The many p-values reported in the abstract, and in the body of the paper, are highly redundant. They are strongly associated with each other because they depend on the same, overlapping, or similar outcomes. Since the trial's actual primary goal was to compare the primary outcome, 3-year DFS between the RDG and LDG arms, results of all these secondary, correlated tests are clutter. Rather than listing p-values, it would be far more useful to report estimated within-arm parameters or parameter differences, along with confidence intervals.

5. It has been well established that using p-values to quantify strength of evidence is poor statistical practice. Providing parameter estimates, or possibly S-values (Rafi and Greenland, *BMC Medical Research Methodology* 2020, 20:244) as refutational evidence, is far more useful. Moreover, since a p-value depends on the assumed model as well as hypotheses, if the Cox model does not fit the data well, then the reported p-values would be even more misleading.

6. Reporting that RDG was found to be both non-inferior and superior to LDG in terms of 3-year DFS rate is redundant. If RDG was superior, it could not have been inferior.

7. Including a column of p-values in Table 1, while a reflection of what is done conventionally, is inappropriate. Randomization does not guarantee covariate balance, and between-arm differences in covariate distributions should be expected. Consequently, the p-values are for multiple tests of hypotheses that make very little sense. See, for example, Senn, *Statistics in Medicine*, 2013, 32(9):1439-50.

8. The figures showing estimates of various cumulative probabilities should be replaced by Kaplan-Meier (KM) estimates of DFS distributions, when these are well-defined, and these KM plots also should include tic marks to show administrative censoring times. Figures 2 and 3 are not KM plot, as stated in their legends. The legend of Figure 4 is inadequate since it fails to identify the subgroups.

9. Figures A, B, C and D all are labeled to be estimates of the same thing. These appear to correspond to 4 different patient subgroups.

10. In Table 2, including fits of Cox models with a single covariate ("univariable analyses") is potentially very misleading because the meaning of the coefficient of a single covariate in such a model assumes that there are no other covariate effects, which in general is not true. Reporting these models only muddies the interpretation of a "multivariable" model that includes all relevant covariates.

Reviewer #3:

Remarks to the Author:

This is a report of a randomized study to show non-inferiority of RDG to LDG for gastric cancer. The authors found that RDG had superior survival than LDG. The results are interesting. However, there are many critical issues in this randomized study and the description in the text.

Major comments

1. This is non-inferiority randomized trial. The authors should describe the statistical hypothesis in the text. So, the primary analysis in the protocol is only confirmation of non-inferiority. So, superiority comparison has not been planned.

2. According to the protocol, the authors set 3-year DFS at 82.3% and non-inferiority margin of 16%. Then, the authors calculated the sample size at 300. I cannot understand non-inferiority margin of 16%, which was extremely large and was not acceptable for a phase III study. Is this a confirmatory phase III study? Or just a randomized phase II study to show minimal efficacy of RDG to LDG? If this is phase II study, please revise the title and the text to show phase II.

3. There is no description on the eligibility and exclusion criteria in the text.

4. There is no definition of MITT in the protocol.
5. The authors described, "DFS and OS were defined as the time from surgery" in the protocol and in the text. This is strange. OS must be calculated from the randomization. This is critical.
6. In the efficacy analysis, the patients who received palliative surgery was excluded. Definition of palliative surgery must be clarified. All 283 patients in the efficacy analysis received R0 surgery? If no, authors should show the number of R0/R1/R2 in each group and discuss on the balance affecting the results.
7. The authors described, "The 3-year DFS rate in the RDG group was 85.8% (95% CI: 80.1%-91.6%), whereas in the LDG group, it was 73.2% (95% CI: 66.0%-80.5%, $p=0.011$). I cannot understand this p value. Is this a test for non-inferiority?"
8. In the LDG group, local recurrence was 7.7% which was extremely high and the most frequent pattern. Authors discussed that RDG can remove deeply located extraperigastric lymph nodes than LDG. However, I do not agree to their opinion. Extent of dissection in gastric cancer surgery was strictly defined by many phase III studies. So, there is no evidence to show dissection efficacy of nodes exceeding D2 area. Moreover, in the previous studies, local recurrence was 2-3% of local recurrence after D2 gastrectomy but around 10% after D1 or D0 gastrectomy. I suppose that the difference of local recurrence was due to low quality of LDG not by due to the difference of R or L. So, the authors should show extent of dissection and number of harvested lymph nodes of each group. Moreover, quality of surgery must be described in detail. Also, more deep discussion is necessary on these factors.
9. The authors stated, "If 147 recurrence was suspected, positron emission tomography/computed tomography scans were conducted to further investigate the condition." This means that CT scan was not prespecified at the visit of every 3 months during the first 2 years and every 6 months thereafter. It is difficult to detect the recurrence only by physical examination or tumor markers. So, the recurrence in this study is not precise.
10. The authors discussed on the relation between low complication and better survival in RDG group. However, the complication rate was similar between the groups. The authors should show the grade of complication of each group.
11. Details of adjuvant chemotherapy (ACT) is lacking. Difference of ACT would also affect the results.
12. The authors described, "many studies have suggested that initiating ACT early is associated with improved survival for patients with GC (34-36). These studies were observational reports showing delayed initiation worsen the prognosis. There is no evidence to show "the earlier, the better". Moreover, the authors did not show the data on the initiation date of ACT in both groups.

Other comments

P1, line 73

"For instance, it only provides a 2D surgical view"

This is incorrect. "Basically" 2D.

P6, line 123

"Surgical quality control was reported previously"

Must be stated in this manuscript.

Table 1

Depth of invasion should be more precisely demonstrated.

Manuscript: NCOMMS-23-51139

Title: Effect of Robotic versus Laparoscopic Distal Gastrectomy on 3-Year Disease-Free Survival among Patients with Resectable Gastric Cancer: Outcomes from a Randomized Controlled Trial

Dear editors and reviewers:

We are grateful to you for your valuable comments and suggestions which help us to improve the quality of the manuscript. We have studied the comments carefully and have made modifications and corrections, which we hope meet your approval. We have revised the manuscript according to your kind advice and the referee's detailed suggestions. Here below is our description on revision.

REVIEWER COMMENTS

Reviewer #1 - Gastric cancer surgery, clinical trials (Remarks to the Author):

The authors report the results of their prospective, randomized, clinical trial evaluating robotic vs laparoscopic gastrectomy for gastric cancer. They are commended for such a large population of a disease site that is difficult to study due to low prevalence in Western countries. While their findings are intriguing and consistent with prior reports favoring robotic surgery for lower surgical morbidity, there are several limitations of the manuscript that render it not acceptable to publication.

1. There is no information regarding the type of chemotherapy patients received postoperatively, how it was tolerated, toxicity, etc. Was chemotherapy prescribed per protocol? Type and cycles of chemotherapy may significantly influence oncologic outcomes, independent of surgical resection modality.

Response: Thank you for your valuable comment. In adherence to the protocol, we routinely recommends chemotherapeutic regimens based on 5-fluorouracil (5-FU) in combination with either platinum-based drugs or docetaxel for patients with pathological stage II or stage III. The above information has been supplemented in the **Methods section** with red mark.

Table S5 presented information regarding postoperative adjuvant chemotherapy for both groups. Aside from the median duration in days between surgery and chemotherapy [RDG vs LDG: 28 vs 32, absolute difference: -4, 95% CI: (-7, -1)], there was no statistical differences between the two groups in terms of chemotherapy cycles and toxicity. The above information has been supplemented in the **Results section** with red mark.

Table S5 Adjuvant Chemotherapy Characteristics of Stage II/III patients by group.

	RDG (n=86)	LDG (n=99)	
	Median (IQR) / N (%)	Median (IQR) / N (%)	p-value
Adjuvant chemotherapy			0.768
Absent	18 (20.9%)	19 (19.2%)	
Present ^a	68 (79.1%)	80 (80.8%)	
Surgical procedure–adjuvant chemotherapy interval, (days)	28 (24-32)	32 (26-42)	0.003
No. of cycles completed, median	6 (3-6)	6 (3-6)	0.795
Cycles of completed ^b			
Cycle 3	55 (80.9%)	63 (78.8%)	0.748
Cycle 4	46 (67.6%)	55 (68.8%)	0.886
Cycle 5	43 (63.2%)	49 (61.3%)	0.804
Cycle 6 or more	41 (60.3%)	45 (56.3%)	0.619
Adverse events ^a			
Grade 1-2	33 (48.5%)	41 (51.3%)	0.741
Grade 3-4	13 (19.1%)	14 (17.5%)	0.800

^a 5-fluorouracil (5-FU) in combination with either platinum-based drugs or docetaxel.

^b For patients with adjuvant chemotherapy.

Abbreviations: LDG, laparoscopic distal gastrectomy; RDG, robotic distal gastrectomy; IQR, interquartile range.

2. There are no data on how many patients underwent D1 vs D2 surgery. There are no information regarding number of lymph nodes that were dissected to evaluate if the oncologic surgery was adequate. There are no information on if postoperative radiation was used in any patients. All of these factors could significantly influence oncologic endpoints.

Response: Thank you for your valuable comment. We divided the lymph node (LN) into 2 regions: perigastric regions (stations 1-6) and extraperigastric regions (stations 7-9, 11p, and 12a)¹. Both of the two regions were within the D2 lymphadenectomy. Lymph node dissection noncompliance was defined as the absence of LNs from more than 1 LN station that should have been excised¹. The above information has been supplemented in the **Methods section** with red mark.

All these patients underwent R0 resection with D2 lymphadenectomy. No patients received postoperative radiotherapy. Generally, the total number of lymph nodes (LNs)

dissected was comparable between the two groups [RDG vs LDG: 40.9 ± 11.2 vs 39.9 ± 12.2 , absolute difference: 1.0, 95% CI: (-1.7-3.8)]. The proportion of patients with over 30 lymph nodes dissected was slightly higher in the RDG group (RDG vs LDG: 85.8% vs 78.2%, absolute difference: 7.6%, 95% CI: (-1.3%, 16.6%)). Further stratified analysis by dividing LNs into perigastric regions (No. 1, 3, 4, 5, 6) and extragastric regions (No. 7, 8a, 9, 11p, 12a) revealed that the number of LNs dissected in extraperigastric regions was significantly higher in the RDG group (RDG vs LDG: 17.6 ± 5.8 vs 15.8 ± 6.6 , absolute difference: 1.8, 95% CI: 0.3-3.2). The comparison of LN noncompliance between the 2 groups indicated that the LN noncompliance rate of the RDG group was significantly lower than that of the LDG group [RDG vs LDG: 24.8% vs 40.1%, absolute difference: -15.3%, 95% CI: (-26.1%, -4.6%)]. The above information has been supplemented in the **Results section** with red mark.

1. Lu J, Zheng CH, Xu BB, et al. Assessment of Robotic Versus Laparoscopic Distal Gastrectomy for Gastric Cancer: A Randomized Controlled Trial. *Ann Surg* 2021; 273(5):858-867.

3. It is not clear to this reviewer why the authors list a meta-analysis, not written at their institution, to report their surgical quality (page 6, reference 9).

Response: Thank you for your valuable comment. The reference was wrong and we have corrected it.

1. Lu J, Zheng CH, Xu BB, et al. Assessment of Robotic Versus Laparoscopic Distal Gastrectomy for Gastric Cancer: A Randomized Controlled Trial. *Ann Surg* 2021; 273(5):858-867.

4. There are no data regarding follow-up compliance.

Response: Thank you for your valuable comment. Patients were followed up until death or for at least 36 months, with the last follow-up conducted on April 30, 2023. The median follow-up period was 48.0 months (95% CI: 45.7-50.3), with a total of 3 patients (1.1%) lost to follow-up (1 in the RDG group and 2 in the LDG group). Except for the patients who were lost to follow-up, all patients were followed up according to the prespecified program. The above information has been supplemented in the **Results section** with red mark.

Reviewer #2 - Biostatistics, clinical trials (Remarks to the Author):

1. There are several published large scale retrospective meta-analyses comparing robotic surgery (RS) to laparoscopic gastric surgery that the introduction does not cite. I have listed 3 papers below. While the types of surgeries and goals of these meta-analyses differ, and these were large meta-analyses rather than a single prospective randomized trial, the results of these studies are highly relevant and should be contrasted with those reported in the present paper. A general result seems to be that robotic surgery takes longer, so a question is whether the trial reported in the present paper is confirming this, or if new or different results are being reported.

Guerrini et al. (International Journal of Surgery, 2020, 82:210-228) considered the comparison for gastric cancer in terms of intra-operative, post-operative, and oncological outcomes.

Leang et al. (Surgery for Obesity and Related Diseases, 2024, 20:62-71) considered bariatric surgery in terms of multiple adverse perioperative outcomes.

Vosberg et al. (Obesity Surgery (2022) 32:2341-2348) considered metabolic and bariatric surgery.

Response: Thank you for your valuable comment. While the three references you've listed are all related to robotic gastric surgery, there are differences between robotic gastric cancer surgery and robotic bariatric surgery.

Two large-scale meta-analyses involving robotic metabolic and bariatric surgery indicated that, compared to laparoscopic surgery, robotic metabolic and bariatric surgery (sleeve gastrectomy, Roux-en-Y gastric bypass) extended the operation time without significantly improving postoperative recovery¹⁻². A large-scale meta-analysis involving more than 17,000 gastric cancer patients also found that the operation time for robotic gastrectomy was significantly longer than laparoscopic surgery, but the intraoperative blood loss and postoperative complication rates were lower in the robotic group³, similar to the early reports of the present study⁴. The long-term oncological results should be the focus of future research^{4,5}. The references has been cited and the above information has been supplemented in the **Discussion section** with red mark.

The differences in intraoperative and short-term postoperative outcomes between the two approaches were thoroughly compared when the secondary endpoints of this study were reported⁴. At present, this paper aims to compare the long-term oncological efficacy of the two surgical methods. This study is first randomized controlled trial (RCT) to compare the long-term oncological effectiveness of these two surgical techniques, and it is hoped to provide high-level evidence for the standardized implementation of robotic gastric cancer surgery.

1. Wesley Vosburg R, Haque O, Roth E. Robotic vs. Laparoscopic Metabolic and Bariatric Surgery, Outcomes over 5 Years in Nearly 800,000 Patients. *Obes Surg* 2022; 32(7):2341-2348.

2. Leang YJ, Mayavel N, Yang WTW, et al. Robotic versus laparoscopic gastric bypass in bariatric surgery: a systematic review and meta-analysis on perioperative outcomes. *Surg Obes Relat Dis* 2024; 20(1):62-71.
3. Guerrini GP, Esposito G, Magistri P, et al. Robotic versus laparoscopic gastrectomy for gastric cancer: The largest meta-analysis. *Int J Surg* 2020; 82:210-228.
4. Lu J, Zheng CH, Xu BB, et al. Assessment of Robotic Versus Laparoscopic Distal Gastrectomy for Gastric Cancer: A Randomized Controlled Trial. *Ann Surg* 2021; 273(5):858-867.
5. Ojima T, Nakamura M, Hayata K, et al. Short-term Outcomes of Robotic Gastrectomy vs Laparoscopic Gastrectomy for Patients With Gastric Cancer: A Randomized Clinical Trial. *JAMA Surg* 2021; 156(10):954-963.

2. The modified ITT sample sizes of 141 and 142 strongly suggest that the randomization was restricted to achieve balance. Was this the case?

Response: Thank you for your valuable comment. This study is a non-inferiority randomized controlled trial with the primary outcome being 3-year DFS. Presuming that the 3-year DFS rate for the LDG group is 82.3%, and setting α at 0.025 and β at 90%, with a margin δ of 16%, we determined that at least 120 patients should be included in each group. Considering a projected dropout rate of 20%, we eventually required a minimum of 150 patients in each group, totaling 300¹.

Following randomization, 7 patients undergoing total gastrectomy and 2 patients who withdrew their informed consent were excluded from the RDG group. In the LDG group, 5 patients undergoing total gastrectomy, 2 unresectable patients, and 1 patient with peritoneal metastasis accompanied by pyloric obstruction undergoing palliative surgery were excluded. Ultimately, 141 patients in the RDG group and 142 patients in the LDG group were included

in the final modified intent-to-treat analysis.

1. Lu J, Zheng CH, Xu BB, et al. Assessment of Robotic Versus Laparoscopic Distal Gastrectomy for Gastric Cancer: A Randomized Controlled Trial. *Ann Surg* 2021; 273(5):858-867.

3. No goodness-of-fit analyses are reported for the Cox model. A potential problem is that, if the proportional hazards assumption was not met by the trial's data, then there would be no single hazard ratio (HR) parameter, and it would become necessary to identify a different regression model that provides an acceptable fit to the data, and report its results. In this case, the currently reported HR values would need to be replaced by appropriate parameter estimates.

Response: Thank you for your valuable comment. Baseline variables that were considered clinically relevant with outcomes were tested of proportional hazards assumption. All the variables satisfied the proportional hazards assumption (**Table S2**). The above information has been supplemented in the **Results section** with red mark.

Table S2 Test of proportional hazards assumption among variables for disease-free survival and overall survival.

	Disease-free Survival			Overall Survival		
	chi-square	df	p value	chi-square	df	p value
Age, ≥65	3.458	1	0.063	0.731	1	0.392
Tumor size, mm	0.325	1	0.569	0.139	1	0.709
Female	0.049	1	0.826	0.875	1	0.350
ECOG PS, score 1	0.054	1	0.816	0.205	1	0.650
Surgical approach, robot vs lap	2.458	1	0.117	2.875	1	0.090
Lymphovascular invasion, yes vs no	0.362	1	0.547	0.763	1	0.382
Histology, undifferentiated	0.336	1	0.562	0.139	1	0.710
pT stage, T2-4 vs T1	0.372	1	0.542	0.372	1	0.542
pN stage, N+ vs N0	0.095	1	0.758	0.019	1	0.891
Adjuvant chemotherapy, yes vs no	1.813	1	0.178	2.439	1	0.118
Global	8.789	10	0.552	8.408	10	0.589

Abbreviations: df, degree of freedom; ECOG PS, Eastern Cooperative Oncology performance status; lap, laparoscopy

4. More generally, the main reason for fitting a regression model in this sort of setting is to assess possible effects of patient baseline covariates on DFS time, in addition to the between-treatment effect. However, no covariate effects are even mentioned in the abstract.

Response: Thank you for your valuable comment. Multivariable Cox regression model including age, tumor size, sex, ECOG PS, lymphovascular invasion, histology, pT stage and pN stage showed that RDG was associated with better 3-year DFS (HR: 0.541; 95% CI: 0.314-0.932). The above information has been supplemented in the **Abstract** with red mark.

5. The many p-values reported in the abstract, and in the body of the paper, are highly redundant. They are strongly associated with each other because they depend on the same, overlapping, or similar outcomes. Since the trial's actual primary goal was to compare the primary outcome, 3-year DFS between the RDG and LDG arms, results of all these secondary, correlated tests are clutter. Rather than listing p-values, it would be far more useful to report estimated within-arm parameters or parameter differences, along with confidence intervals.

Response: Thank you for your valuable comment. According to your comments, we have deleted the p-values and reported the estimated within-arm parameters or parameter differences, along with confidence intervals, for each outcome in our study.

6. It has been well established that using p-values to quantify strength of evidence is poor statistical practice. Providing parameter estimates, or possibly S-values (Rafi and Greenland, BMC Medical Research Methodology 2020, 20:244) as refutational evidence, is far more useful. Moreover, since a p-value depends on the assumed model as well as hypotheses, if the

Cox model does not fit the data well, then the reported p-values would be even more misleading.

Response: Thank you for your valuable comment. According to your comments, we have reported the estimated within-arm parameters or parameter differences, along with confidence intervals, for each outcome in our study. In addition, baseline variables that were considered clinically relevant with outcomes were tested of proportional hazards assumption. All the variables satisfied the proportional hazards assumption (**Table S2**).

7. Reporting that RDG was found to be both non-inferior and superior to LDG in terms of 3-year DFS rate is redundant. If RDG was superior, it could not have been inferior.

Response: Thank you for your valuable comment. Given the non-inferiority design of this study and the achievement of the primary endpoint, we have revised the descriptions into "Compared to LDG, RDG demonstrated non-inferiority in the 3-year DFS rate."

8. Including a column of p-values in Table 1, while a reflection of what is done conventionally, is inappropriate. Randomization does not guarantee covariate balance, and between-arm differences in covariate distributions should be expected. Consequently, the p-values are for multiple tests of hypotheses that make very little sense. See, for example, Senn, *Statistics in Medicine*, 2013, 32(9):1439-50.

Response: Thank you for your valuable comment. The p-values in Table 1 have been deleted.

9. The figures showing estimates of various cumulative probabilities should be replaced by

Kaplan-Meier (KM) estimates of DFS distributions, when these are well-defined, and these KM plots also should include tic marks to show administrative censoring times. Figures 2 and 3 are not KM plot, as stated in their legends. The legend of Figure 4 is inadequate since it fails to identify the subgroups.

Response: Thank you for your valuable comment. The Figures 2 and 3 have been replaced with Kaplan-Meier plots as suggested. Additionally, the legend of Figure 4 has been revised into “Subgroup analysis of disease-free survival including age, sex, ECOG PS, lymphovascular invasion, histology, tumor size, postoperative complication and adjuvant chemotherapy”.

10. Figures A, B, C and D all are labeled to be estimates of the same thing. These appear to correspond to 4 different patient subgroups.

Response: Thank you for your valuable comment. Figure 3 showed the Kaplan-Meier curves of disease-free survival for robotic and laparoscopic distal gastrectomies within 3 years after surgery by different pathologic T stage or N stage. The four contents of A, B, C, and D were as follows: (A) only patients with pT1 stage; (B) only patients with pT2-4 stage; (C) only patients with pN0 stage; (D) only patients with pN+ stage. We have identified them in the figures to make the results clearer.

11. In Table 2, including fits of Cox models with a single covariate (“univariable analyses”) is potentially very misleading because the meaning of the coefficient of a single covariate in such a model assumes that there are no other covariate effects, which in general is not true.

Reporting these models only muddies the interpretation of a “multivariable” model that includes all relevant covariates.

Response: Thank you for your valuable comment. Baseline variables that were considered clinically relevant with outcomes were tested of proportional hazards assumption and then entered into the multivariate Cox proportional hazards model (forward stepwise). The above information has been supplemented in the **Methods section** with red mark.

Multivariable Cox regression model including age, tumor size, sex, ECOG PS, lymphovascular invasion, histology, pT stage and pN stage showed that RDG remained an independent protective factor for the 3-year DFS (HR: 0.541; 95% CI: 0.314-0.932). The above information has been supplemented in the **Results section** with red mark.

Reviewer #3 - Gastric cancer surgery, clinical trials (Remarks to the Author):

This is a report of a randomized study to show non-inferiority of RDG to LDG for gastric cancer. The authors found that RDG had superior survival than LDG. The results are interesting. However, there are many critical issues in this randomized study and the description in the text.

Major comments

1. This is non-inferiority randomized trial. The authors should describe the statistical hypothesis in the text. So, the primary analysis in the protocol is only confirmation of non-inferiority. So, superiority comparison has not been planned.

Response: Thank you for your valuable comment. The 3-year DFS rate in the RDG group

was 85.8% (95% CI: 80.1%-91.6%), whereas in the LDG group, it was 73.2% (95% CI: 66.0%-80.5%). The absolute between-group difference was 12.6% (95% CI: 3.3%-21.9%), that did not cross the prespecified non-inferiority margin of -16%, thus satisfying the primary hypothesis of non-inferiority of the RDG as compared with the LDG at 36-month follow-up. Furthermore, given that the two-sided 95% CI lies wholly to the right of zero, the RDG might be even superior to LDG, although there is a need for specific confirmatory superiority trials¹. The above information has been supplemented in the **Results section** with red mark.

Our findings suggested that RDG could be even more efficacious than the LDG as regard to the 3-year DFS and local recurrence rate. However, the past evidence suggested us to design this study as a non-inferiority trial. Several RCTs with a non-inferiority design also revealed the similar situation²⁻⁴. As a consequence, superiority of the RDG should be verified with a specifically designed superiority trial and should be thus object of future investigations. A post hoc calculation was performed based on our findings and, assuming a superiority margin of 12.6%, with a 3-year DFS of 73.2% and an expected rate in the experimental group of at least 85.6%, 161 per group would be needed, with an 90% power and alpha at 0.025. Given the proximity to the calculated sample size, therefore, the results of this study, which portends towards a potential of superiority of RDG, is of great importance in clinical practice and suggest that we design a superiority trial based on the evidence emerged from the present study. The above information has been supplemented in the **Discussion section** with red mark.

1. Piaggio G, Elbourne DR, Pocock SJ, et al. Reporting of noninferiority and equivalence randomized trials: extension of the CONSORT 2010 statement. *Jama* 2012; 308(24):2594-604.

2. Familiari P, Borrelli de Andreis F, Landi R, et al. Long versus short peroral endoscopic myotomy for the treatment of achalasia: results of a non-inferiority randomised controlled trial. *Gut* 2023; 72(8):1442-1450.
3. Philis-Tsimikas A, Asong M, Franek E, et al. Switching to once-weekly insulin icodec versus once-daily insulin degludec in individuals with basal insulin-treated type 2 diabetes (ONWARDS 2): a phase 3a, randomised, open label, multicentre, treat-to-target trial. *Lancet Diabetes Endocrinol* 2023; 11(6):414-425.
4. Gao L, Lee BW, Chawla M, et al. Tirzepatide versus insulin glargine as second-line or third-line therapy in type 2 diabetes in the Asia-Pacific region: the SURPASS-AP-Combo trial. *Nat Med* 2023; 29(6):1500-1510.

2. According to the protocol, the authors set 3-year DFS at 82.3% and non-inferiority margin of 16%. Then, the authors calculated the sample size at 300. I cannot understand non-inferiority margin of 16%, which was extremely large and was not acceptable for a phase III study. Is this a confirmatory phase III study? Or just a randomized phase II study to show minimal efficacy of RDG to LDG? If this is phase II study, please revise the title and the text to show phase II.

Response: Thank you for your valuable comment. As you commented, this study is a phase II randomized controlled study. We have supplemented the corresponding information in **Title** and **Methods section** with red mark.

3. There is no description on the eligibility and exclusion criteria in the text.

Response: Thank you for your valuable comment.

The inclusion criteria for this study are as follows: (1) Age from 18 to 75 years (not including 18 and 75 years old); (2) Primary gastric adenocarcinoma confirmed pathologically by

endoscopic biopsy; (3) Clinical stage tumor T1-4a (cT1-4a), N0/+, M0 at preoperative evaluation according to the American Joint Committee on Cancer (AJCC) Cancer Staging Manual Eighth Edition; (4) Expected to undergo distal gastrectomy and D1+/D2 lymph node dissection to obtain R0 surgical results; (5) Performance status of 0 or 1 on the ECOG (Eastern Cooperative Oncology Group) scale; (6) ASA class I to III; (7) Written informed consent.

The exclusion criteria for this study are as follows: (1) Women during pregnancy or breast-feeding; (2) Severe mental disorder; (3) History of previous upper abdominal surgery (except for laparoscopic cholecystectomy); (4) History of previous gastric surgery (including ESD/EMR for gastric cancer); (5) Multiple primary gastric cancer; (6) Enlarged or bulky regional lymph node diameter over 3cm by preoperative imaging; (7) History of other malignant disease within past five years; (8) History of previous neoadjuvant chemotherapy or radiotherapy; (9) History of unstable angina or myocardial infarction within the past six months; (10) History of cerebrovascular accident within past six months; (11) History of continuous systematic administration of corticosteroids within one month; (12) Requirement of simultaneous surgery for another disease; (13) Emergency surgery due to complications (bleeding, obstruction or perforation) caused by gastric cancer; (14) Forced expiratory volume in 1 second (FEV1) < 50% of the predicted values.

The above information has been supplemented in the **Methods** section with red mark.

4. There is no definition of MITT in the protocol.

Response: Thank you for your valuable comment. The modified intent-to-treat (MITT) group

excluded patients who met exclusion criteria post-randomization. These criteria, as defined in the study protocol, included intraoperatively or postoperatively confirmed T4b or M1, unresectable tumors, total gastrectomy, and withdrawal of informed consent. The above information has been supplemented in the **Methods** section with red mark.

5. The authors described, “DFS and OS were defined as the time from surgery” in the protocol and in the text. This is strange. OS must be calculated from the randomization. This is critical.

Response: Thank you for your valuable comment. As the reviewer mentioned, in studies where patients were treated with both perioperative drug therapy and surgery, researchers typically define the start of survival time calculation as the time of randomization¹⁻². Given that in studies involving only surgical treatments, researchers aim to compare the efficacy of different surgical interventions, and the time from randomization to surgical intervention is typically around two days, this study, similar to previous RCTs³⁻⁵, defines the start of survival time calculation as the time when surgery begins.

1. Zhang X, Liang H, Li Z, et al. Perioperative or postoperative adjuvant oxaliplatin with S-1 versus adjuvant oxaliplatin with capecitabine in patients with locally advanced gastric or gastro-oesophageal junction adenocarcinoma undergoing D2 gastrectomy (RESOLVE): an open-label, superiority and non-inferiority, phase 3 randomised controlled trial. *Lancet Oncol* 2021; 22(8):1081-1092.

2. Janjigian YY, Shitara K, Moehler M, et al. First-line nivolumab plus chemotherapy versus chemotherapy alone for advanced gastric, gastro-oesophageal junction, and oesophageal adenocarcinoma (CheckMate 649): a randomised, open-label, phase 3 trial. *Lancet* 2021; 398(10294):27-40.

3. Yu J, Huang C, Sun Y, et al. Effect of Laparoscopic vs Open Distal Gastrectomy on 3-Year Disease-Free Survival in Patients With Locally Advanced Gastric Cancer: The CLASS-01 Randomized Clinical Trial. *JAMA* 2019; 321(20):1983-1992.
4. Hyung WJ, Yang HK, Park YK, et al. Long-Term Outcomes of Laparoscopic Distal Gastrectomy for Locally Advanced Gastric Cancer: The KLASS-02-RCT Randomized Clinical Trial. *J Clin Oncol* 2020; 38(28):3304-3313.
5. Chen QY, Zhong Q, Liu ZY, et al. Indocyanine green fluorescence imaging-guided versus conventional laparoscopic lymphadenectomy for gastric cancer: long-term outcomes of a phase 3 randomised clinical trial. *Nat Commun* 2023; 14(1):7413.

6. In the efficacy analysis, the patients who received palliative surgery was excluded. Definition of palliative surgery must be clarified. All 283 patients in the efficacy analysis received R0 surgery? If no, authors should show the number of R0/R1/R2 in each group and discuss on the balance affecting the results.

Response: Thank you for your valuable comment. Palliative surgery is defined as non-curative gastrectomy or gastrojejunostomy performed to alleviate severe complications (such as bleeding or obstruction) caused by the tumor in patients with advanced or metastatic gastric cancer. The above information has been supplemented in the **Methods section** with red mark.

In the LDG group, there was a case of gastric cancer with pyloric obstruction. Intraoperatively, peritoneal metastasis was observed, and palliative gastrectomy was performed to relieve the obstruction symptoms. In the MITT analysis, we excluded this patient. All patients included in the final analysis underwent R0 resection. The above information has been supplemented in

the **Results section** with red mark.

7. The authors described, “The 3-year DFS rate in the RDG group was 85.8% (95% CI: 80.1%-91.6%), whereas in the LDG group, it was 73.2% (95% CI: 66.0%-80.5%, $p=0.011$). I cannot understand this p value. Is this a test for non-inferiority?”

Response: Thank you for your valuable comment. The p-value you referenced is derived from the log-rank test and it is not a test for non-inferiority. According to the comments from **Reviewer 2**, we have deleted the p value and reported the estimated within-arm parameters or parameter differences, along with confidence intervals, for each outcome in our study. The 3-year DFS rate in the RDG group was 85.8% (95% CI: 80.2%-91.8%), whereas in the LDG group, it was 73.2% (95% CI: 66.3%-80.9%). The absolute between-group difference (AD) was 12.6% (95% CI: 3.3%-21.9%).

8. In the LDG group, local recurrence was 7.7% which was extremely high and the most frequent pattern. Authors discussed that RDG can remove deeply located extraperigastric lymph nodes than LDG. However, I do not agree to their opinion. Extent of dissection in gastric cancer surgery was strictly defined by many phase III studies. So, there is no evidence to show dissection efficacy of nodes exceeding D2 area. Moreover, in the previous studies, local recurrence was 2-3% of local recurrence after D2 gastrectomy but around 10% after D1 or D0 gastrectomy. I suppose that the difference of local recurrence was due to low quality of LDG not by due to the difference of R or L. So, the authors should show extent of dissection and number of harvested lymph nodes of each group. Moreover, quality of surgery must be

described in detail. Also, more deep discussion is necessary on these factors.

Response: Thank you for your valuable comment. We divided the lymph node (LN) into 2 regions: perigastric regions (stations 1-6) and extraperigastric regions (stations 7-9, 11p, and 12a)¹. Both of the two regions were within the D2 lymphadenectomy. Lymph node dissection noncompliance was defined as the absence of LNs from more than 1 LN station that should have been excised¹. The above information has been supplemented in the **Methods section** with red mark.

All the patients underwent R0 resection with D2 lymphadenectomy. Generally, the total number of lymph nodes (LNs) dissected was comparable between the two groups [RDG vs LDG: 40.9 ± 11.2 vs 39.9 ± 12.2 , absolute difference: 1.0, 95% CI: (-1.7-3.8)]. The proportion of patients with over 30 lymph nodes dissected was slightly higher in the RDG group (RDG vs LDG: 85.8% vs 78.2%, absolute difference: 7.6%, 95% CI: (-1.3%, 16.6%)). Further stratified analysis by dividing LNs into perigastric regions (No. 1, 3, 4, 5, 6) and extragastric regions (No. 7, 8a, 9, 11p, 12a) revealed that the number of LNs dissected in extraperigastric regions was significantly higher in the RDG group (RDG vs LDG: 17.6 ± 5.8 vs 15.8 ± 6.6 , absolute difference: 1.8, 95% CI: 0.3-3.2). The comparison of LN noncompliance between the 2 groups indicated that the LN noncompliance rate of the RDG group was significantly lower than that of the LDG group [RDG vs LDG: 24.8% vs 40.1%, absolute difference: -15.3%, 95% CI: (-26.1%, -4.6%)]. The above information has been supplemented in the **Results section** with red mark.

Previous studies indicated that the incidence of local recurrence in patients undergoing D2 curative gastrectomy varies from 3% to 20%²⁻¹¹. In studies involving postoperative

recurrence rates, there are two methods of enumeration. One approach involves repeated counting. As some patients have recurrences in different locations at the time of initial recurrence detection (i.e., multiple sites), these are counted repeatedly to provide a more comprehensive depiction of recurrence types²⁻⁶. This is the method of demonstrating recurrence patterns that we used in the present study, as shown in the Venn diagram (**Fig. S4**). Another instance involves counting patients with simultaneous recurrences in different locations as a singular type of recurrence, without repeating the count with other types⁷⁻¹¹. Different methods of enumeration lead to variations in recurrence patterns, and in several studies including CLASS 01, local recurrence remains the most common type of recurrence^{8,9,11}. When repeated counting is not taken into account, the local recurrence rate in the present study is 4.9% (7/142) (**Fig. S4**), which is similar with previous studies⁸⁻¹¹. The above information has been supplemented in the **Discussion section** with red mark.

1. Lu J, Zheng CH, Xu BB, et al. Assessment of Robotic Versus Laparoscopic Distal Gastrectomy for Gastric Cancer: A Randomized Controlled Trial. *Ann Surg* 2021; 273(5):858-867.
2. Hyung WJ, Yang HK, Park YK, et al. Long-Term Outcomes of Laparoscopic Distal Gastrectomy for Locally Advanced Gastric Cancer: The KLASS-02-RCT Randomized Clinical Trial. *J Clin Oncol* 2020; 38(28):3304-3313.
3. Lin JX, Lin JP, Wang ZK, et al. Assessment of Laparoscopic Spleen-Preserving Hilar Lymphadenectomy for Advanced Proximal Gastric Cancer Without Invasion Into the Greater Curvature: A Randomized Clinical Trial. *JAMA Surg* 2023; 158(1):10-18.
4. Bang Y-J, Kim Y-W, Yang H-K, et al. Adjuvant capecitabine and oxaliplatin for gastric cancer after D2 gastrectomy (CLASSIC): a phase 3 open-label, randomised controlled trial. *The Lancet* 2012; 379(9813):315-321.

5. Chen QY, Zhong Q, Liu ZY, et al. Does Noncompliance in Lymph Node Dissection Affect Oncological Efficacy in Gastric Cancer Patients Undergoing Radical Gastrectomy? *Ann Surg Oncol* 2019; 26(6):1759-1771.
6. Lin JX, Lin JP, Xie JW, et al. Prognostic importance of the preoperative modified systemic inflammation score for patients with gastric cancer. *Gastric Cancer* 2019; 22(2):403-412.
7. Lee JH, Chang KK, Yoon C, et al. Lauren Histologic Type Is the Most Important Factor Associated With Pattern of Recurrence Following Resection of Gastric Adenocarcinoma. *Ann Surg* 2018; 267(1):105-113.
8. Li ZY, Zhou YB, Li TY, et al. Robotic Gastrectomy Versus Laparoscopic Gastrectomy for Gastric Cancer: A Multicenter Cohort Study of 5402 Patients in China. *Ann Surg* 2023; 277(1):e87-e95.
9. Yu J, Huang C, Sun Y, et al. Effect of Laparoscopic vs Open Distal Gastrectomy on 3-Year Disease-Free Survival in Patients With Locally Advanced Gastric Cancer: The CLASS-01 Randomized Clinical Trial. *JAMA* 2019; 321(20):1983-1992.
10. Chen QY, Zhong Q, Liu ZY, et al. Indocyanine green fluorescence imaging-guided versus conventional laparoscopic lymphadenectomy for gastric cancer: long-term outcomes of a phase 3 randomised clinical trial. *Nat Commun* 2023; 14(1):7413.
11. Li ZY, Wei B, Zhou YB, et al. Long-term oncological outcomes of robotic versus laparoscopic gastrectomy for gastric cancer: multicentre cohort study. *Br J Surg* 2024; 111(1).

9. The authors stated, "If recurrence was suspected, positron emission tomography/computed tomography scans were conducted to further investigate the condition." This means that CT scan was not prespecified at the visit of every 3 months during the first 2 years and every 6 months thereafter. It is difficult to detect the recurrence only by physical examination or tumor markers. So, the recurrence in this study is not precise.

Response: Thank you for your valuable comment. As the reviewer pointed out, under existing medical conditions, it is challenging to detect early recurrence accurately in patients after radical gastrectomy. In many previous large-scale RCTs, similar follow-up strategies were adopted for recurrence detection as in this study^{1,2}.

As we have described in the **Methods section**, the follow-up strategy for this study is as follows: At the visit of every 3 months during the first 2 years and every 6 months thereafter, follow-up assessments encompassed various components including physical examinations, laboratory tests (peripheral blood routine assessment, blood biochemistry, serum tumor markers including CA19-9, CA72-4, and CEA level), chest radiography, **abdominal computed tomographic scans**, and annual endoscopic examinations. When the above examinations indicates a possible recurrence, positron emission tomography/computed tomography scans were conducted to further investigate the condition. We revised the sentence you mentioned into “Should a recurrence be suspected during the aforementioned examinations, positron emission tomography/computed tomography scans were conducted to further investigate the condition” for enhanced clarity with red mark .

1. Yu J, Huang C, Sun Y, et al. Effect of Laparoscopic vs Open Distal Gastrectomy on 3-Year Disease-Free Survival in Patients With Locally Advanced Gastric Cancer: The CLASS-01 Randomized Clinical Trial. *JAMA* 2019; 321(20):1983-1992.

2. Hyung WJ, Yang HK, Park YK, et al. Long-Term Outcomes of Laparoscopic Distal Gastrectomy for Locally Advanced Gastric Cancer: The KLASS-02-RCT Randomized Clinical Trial. *J Clin Oncol* 2020; 38(28):3304-3313.

10. The authors discussed on the relation between low complication and better survival in

RDG group. However, the complication rate was similar between the groups. The authors should show the grade of complication of each group.

Response: Thank you for your valuable comment. Previous reports on the short-term outcomes of this study indicated that the overall postoperative complication rate was significantly lower in the RDG group than in the LDG group (RDG vs LDG: 9.2% vs 17.6%, absolute difference: -8.4%, 95% CI: -16.3% to -0.5%). Based on the ClavienDindo classification system, significant difference was observed in the grade II complications between the 2 groups (RDG vs LDG: 7.8% vs 15.5%, absolute difference: -7.7%, 95% CI: -15.1% to -0.3%). The above information has been supplemented in the **Discussion section** with red mark.

1. Lu J, Zheng CH, Xu BB, et al. Assessment of Robotic Versus Laparoscopic Distal Gastrectomy for Gastric Cancer: A Randomized Controlled Trial. *Ann Surg* 2021; 273(5):858-867.

11. Details of adjuvant chemotherapy (ACT) is lacking. Difference of ACT would also affect the results.

Response: Thank you for your valuable comment. **Table S5** presented information regarding postoperative adjuvant chemotherapy for both groups. Aside from the median duration in days between surgery and chemotherapy [RDG vs LDG: 28 vs 32, absolute difference: -4, 95% CI: (-7, -1)], there was no statistical differences between the two groups in terms of chemotherapy cycles and toxicity. The above information has been supplemented in the **Results section** with red mark.

Table S5 Adjuvant Chemotherapy Characteristics of Stage II/III patients by group.

	RDG (n=86)	LDG (n=99)	
	Median (IQR) / N (%)	Median (IQR) / N (%)	p-value
Adjuvant chemotherapy			0.768
Absent	18 (20.9%)	19 (19.2%)	
Present ^a	68 (79.1%)	80 (80.8%)	
Surgical procedure–adjuvant chemotherapy interval, (days)	28 (24-32)	32 (26-42)	0.003
No. of cycles completed, median	6 (3-6)	6 (3-6)	0.795
Cycles of completed ^b			
Cycle 3	55 (80.9%)	63 (78.8%)	0.748
Cycle 4	46 (67.6%)	55 (68.8%)	0.886
Cycle 5	43 (63.2%)	49 (61.3%)	0.804
Cycle 6 or more	41 (60.3%)	45 (56.3%)	0.619
Adverse events ^a			
Grade 1-2	33 (48.5%)	41 (51.3%)	0.741
Grade 3-4	13 (19.1%)	14 (17.5%)	0.800

^a 5-fluorouracil (5-FU) in combination with either platinum-based drugs or docetaxel.

^b For patients with adjuvant chemotherapy.

Abbreviations: LDG, laparoscopic distal gastrectomy; RDG, robotic distal gastrectomy; IQR, interquartile range.

12. The authors described, “many studies have suggested that initiating ACT early is associated with improved survival for patients with GC (34-36). These studies were observational reports showing delayed initiation worsen the prognosis. There is no evidence to show “the earlier, the better”. Moreover, the authors did not show the data on the initiation date of ACT in both groups.

Response: Thank you for your valuable comment. As the reviewer commented, there is no evidence to show “the earlier, the better” . We revised the sentence into “many studies have suggested that initiating ACT late is associated with worse survival for patients with GC” with red mark.

Table S5 presented information regarding postoperative adjuvant chemotherapy for both groups. Aside from the median duration in days between surgery and chemotherapy [RDG vs LDG: 28 vs 32, absolute difference: -4, 95% CI: (-7, -1)], there was no statistical differences

between the two groups in terms of chemotherapy cycles and toxicity. The above information has been supplemented in the **Results section** with red mark.

Other comments

1. P1, line 73, “For instance, it only provides a 2D surgical view”

This is incorrect. “Basically” 2D.

Response: Thank you for your valuable comment. We revised the sentence into “For instance, it only provides a **basically** 2D surgical view” with red mark.

2. P6, line 123, “Surgical quality control was reported previously”

Must be stated in this manuscript.

Response: Thank you for your valuable comment. The Da Vinci robotic system (Intuitive Surgical, Inc, Sunnyvale, CA) was used to perform all the RGs by the same group of surgeons (CM.H and CH.Z) with experience of more than 300 laparoscopic and 50 robotic operations for GC before joining the trial¹. According to the 2014 Japanese Gastric Cancer Treatment Guidelines, lymphadenectomy includes No. 1, 3, 4sb, 4d, 5, 6, 7, 8a, 9, 11p, and 12a (D2)². Another group of surgeons reviewed the unedited videos of participants’ lymphadenectomies once a week using a sample survey (**Table S1**) for standardization and quality control^{1,3}. All the D2 gastrectomies were determined acceptable. The above descriptions have been added in the **Methods section** with red mark.

1. Lu J, Zheng CH, Xu BB, et al. Assessment of Robotic Versus Laparoscopic Distal Gastrectomy for Gastric Cancer: A Randomized Controlled Trial. *Ann Surg* 2021; 273(5):858-867.
2. Japanese Gastric Cancer A. Japanese gastric cancer treatment guidelines 2014 (ver. 4). *Gastric*

Table S1. Checklist for the determining the success of D2 lymphadenectomy

Scoring Method for D2 Lymph Node Dissection	Complete Incomplete None		
	10	5	0
1. Properly full omentectomy	[ ]	[ ]	[ ]
2. Ligation of left gastroepiploic artery at origin	[ ]	[ ]	[ ]
3. Ligation of right gastroepiploic artery at origin	[ ]	[ ]	[ ]
4. Full exposure of common hepatic artery	[ ]	[ ]	[ ]
5. Ligation of right gastric artery at origin	[ ]	[ ]	[ ]
6. Exposure of portal vein	[ ]	[ ]	[ ]
7. Exposure of splenic artery to branch of posterior gastric artery	[ ]	[ ]	[ ]
8. Identification of splenic vein	[ ]	[ ]	[ ]
9. Ligation of left gastric artery at origin	[ ]	[ ]	[ ]
10. Exposure of gastroesophageal junction	[ ]	[ ]	[ ]

1. Properly full omentectomy
 - a. Omentectomy was performed close to transverse colon
 - b. Omentectomy was performed from hepatic flexure to splenic flexure
 - c. Anterior layer of transverse colonic mesentery and pancreatic anterior peritoneum was dissected.
2. Ligation of left gastroepiploic artery at origin
3. Ligation of right gastroepiploic artery at origin
4. Full exposure of common hepatic artery
 - a. More than half of anterior part in the common hepatic artery were exposed.
5. Ligation of right gastric artery at origin
6. Exposure of portal vein
7. Exposure of splenic artery to branch of posterior gastric artery
 - a. More than half of anterior part in splenic artery was exposed.
 - b. Splenic artery was exposed from celiac trunk to posterior gastric artery
8. Identification of splenic vein
9. Ligation of left gastric artery at origin
10. Exposure of gastroesophageal junction

- a. Anterior and right side of the abdominal esophagus were exposed.
- D2 lymphadenectomy was accepted if all randomly assigned three investigators rated 85 points and more regarding checklists in unedited video review.

3. Table 1, Depth of invasion should be more precisely demonstrated.

Response: Thank you for your valuable comment. We have divided pT stage into pT1, pT2, pT3 and pT4a stages in Table 1 with red mark.

In conclusion, we have checked the manuscript and revised it according to all the comments. We submit here the revised manuscript as well as a list of changes. If you have any question about this manuscript, please don't hesitate to let me know.

Sincerely yours,

Prof. Chang-Ming Huang

Department of Gastric Surgery, Fujian Medical University Union Hospital,

E-mail: hcmlr2002@163.com

Reviewers' Comments:

Reviewer #1:

Remarks to the Author:

The authors have addressed many of the reviewer concerns. However, it is still not clear why they are presenting their primary endpoint as superiority fashion when it was a non-inferiority trial. The chemotherapy details provided are not sufficient to understand if one group received more docetaxel or oxaliplatin. Appreciate the concerns raised by statistical reviewer.

Reviewer #2:

Remarks to the Author:

Referee's report on *Nature Communications* manuscript NCOMMS-23-51139A, the first revision of "Effect of Robotic versus Laparoscopic Distal Gastrectomy on 3-Year Disease-Free Survival among Patients with Resectable Gastric Cancer: Outcomes from a Randomized Controlled Trial" by Lu et al.

Comments for the Authors on the Revision

1. The authors have done a creditable job of responding to my comments on the first draft. The following remaining methodological issues should be addressed.
2. Tables 2 and 3. Displaying p-values $< .05$ in red is inappropriate. There is nothing sacred about the conventional cut-off $.05$, and its common use to represent "significance" is bad scientific practice. The interpretation of strength of evidence for refuting a null hypothesis provided by a p-value depends on sample size, parameter estimates, and model assumptions. For example, the estimated HR for tumor size in Table 1 of 1.014, with 95% CI 1.000 – 1.028, is trivially different from 1, despite the fact that the p-value 0.048 is nominally significant. In fact, the p-value = .048 gives S-value = 4 which, as refutational evidence for the null hypothesis that this HR = 1, is as surprising as flipping a coin that one believes is fair 4 times and observing 4 heads.
3. Reporting p-values computed in two different ways in Table 3 is redundant and a distraction. In any case, the distribution theory underlying the chi-square tests is asymptotic, which renders the numerical p-values unreliable given the small event counts.
4. In general, if multiple p-values are reported in an analysis, they should be adjusted upward to correct for multiple testing to control the overall false positive rate. In each table that gives multiple p-values, these should be replaced by upwardly adjusted values. There are many methods to adjust for multiple testing (Bonferroni, Holm, etc.). See, for example, Chen et al. *J Thoracic Disease*, Vol 9, No 6, 2017.
The need to upwardly adjust p-values for multiple tests is due to the fact that the overall false positive rate is inflated. For example, if 20 tests are performed, the null probability that at least one of the 20 tests will have p-value $< .05$ equals $.64$. Consequently, when multiple p-values are presented, reporting unadjusted values is not correct and is misleading.
5. The table of subgroup-treatment interaction HR estimates giving a forest plot of 17 different confidence intervals, also suffers from the problem of multiple testing. Only adjusted p-values should be reported there.

6. It still appears to me that the randomization was restricted to achieve balance, as shown by the figure in which 300 patients were randomized with 150 to LADG and 150 to RADG. This sample size outcome is extremely unlikely if a simple fair randomization was performed independently for each patient at enrollment. This is not a major point, but I find the sample sizes puzzling.

Reviewer #3:

Remarks to the Author:

The authors appropriately responded to the reviewer's comments.

Manuscript: NCOMMS-23-51139A

Title: Effect of Robotic versus Laparoscopic Distal Gastrectomy on 3-Year Disease-Free Survival among Patients with Resectable Gastric Cancer: Outcomes from a Phase 2 Randomized Controlled Trial

Dear editors and reviewers:

We are grateful to you for your valuable comments and suggestions which help us to improve the quality of the manuscript. We have studied the comments carefully and have made modifications and corrections, which we hope meet your approval. We have revised the manuscript according to your kind advice and the referee's detailed suggestions. Here below is our description on revision.

REVIEWER COMMENTS

Reviewer #1 (Remarks to the Author):

1. The authors have addressed many of the reviewer concerns. However, it is still not clear why they are presenting their primary endpoint as superiority fashion when it was a non-inferiority trial.

Response: Thank you for your valuable comment. As you commented, this is a non-inferiority trial. The primary analysis in the protocol is only confirmation of non-inferiority. So, superiority comparison has not been planned. Similarly, several RCTs with a non-inferiority design also found potential therapeutic advantages in support of the the experimental group¹⁻³. According to your comments, we revised the the relevant description

into a comparison of non-inferiority. The detailed information is as follows.

The 3-year DFS rate in the RDG group was 85.8% (95% CI: 80.1%-91.6%), whereas in the LDG group, it was 73.2% (95% CI: 66.0%-80.5%). The absolute between-group difference was 12.6% (95% CI: 3.3%-21.9%), that did not cross the prespecified non-inferiority margin of -16%, thus satisfying the primary hypothesis of non-inferiority of the RDG as compared with the LDG at 36-month follow-up. The above information has been supplemented in the **Results section** with red mark.

In addition, we revised the conclusion into “Compared to LDG, RDG demonstrated non-inferiority in the 3-year DFS rate” with red mark.

1. Familiari P, Borrelli de Andreis F, Landi R, et al. Long versus short peroral endoscopic myotomy for the treatment of achalasia: results of a non-inferiority randomised controlled trial. *Gut* 2023; 72(8):1442-1450.
2. Philis-Tsimikas A, Asong M, Franek E, et al. Switching to once-weekly insulin icodec versus once-daily insulin degludec in individuals with basal insulin-treated type 2 diabetes (ONWARDS 2): a phase 3a, randomised, open label, multicentre, treat-to-target trial. *Lancet Diabetes Endocrinol* 2023; 11(6):414-425.
3. Gao L, Lee BW, Chawla M, et al. Tirzepatide versus insulin glargine as second-line or third-line therapy in type 2 diabetes in the Asia-Pacific region: the SURPASS-AP-Combo trial. *Nat Med* 2023; 29(6):1500-1510.

2. The chemotherapy details provided are not sufficient to understand if one group received more docetaxel or oxaliplatin.

Response: Thank you for your valuable comment. We have supplemented the differences in the specific chemotherapy regimen of the two groups in **Table S5**. Seventy-three patients

(91.25%) in the LDG group received docetaxel based chemotherapy regimen, compared to 61 patients (89.7%) in the RDG group (absolute difference: 1.5%, 95% CI: -8.0% to 11.1%). The above information has been supplemented in the **Results section** with red mark.

Table S5 Adjuvant Chemotherapy Characteristics of Stage II/III patients by group.			
	RDG (n=86)	LDG (n=99)	
	Median (IQR) / N (%)	Median (IQR) / N (%)	p- value
Adjuvant chemotherapy			0.768
Absent	18 (20.9%)	19 (19.2%)	
Present a	68 (79.1%)	80 (80.8%)	
Chemotherapy regimens b			0.749
Platinum based	7 (10.3%)	7 (8.75%)	
Docetaxel based	61 (89.7%)	73 (91.25%)	
Surgical procedure–adjuvant chemotherapy interval, (c	28 (24-32)	32 (26-42)	0.003
No. of cycles completed, median	6 (3-6)	6 (3-6)	0.795
Cycles of completed b			
Cycle 3	55 (80.9%)	63 (78.8%)	0.748
Cycle 4	46 (67.6%)	55 (68.8%)	0.886
Cycle 5	43 (63.2%)	49 (61.3%)	0.804
Cycle 6 or more	41 (60.3%)	45 (56.3%)	0.619
Adverse events			
Grade 1-2	33 (48.5%)	41 (51.3%)	0.741
Grade 3-4	13 (19.1%)	14 (17.5%)	0.800

a 5-fluorouracil (5-FU) in combination with either platinum-based drugs or docetaxel.
b For patients with adjuvant chemotherapy.
Abbreviations: LDG, laparoscopic distal gastrectomy; RDG, robotic distal gastrectomy; IQR, interquartile range.

3. Appreciate the concerns raised by statistical reviewer.

Response: Thanks for the reviewers' valuable comments and suggestions which help us to improve the quality of the manuscript.

Reviewer #2 (Remarks to the Author):

1. The authors have done a creditable job of responding to my comments on the first draft.

The following remaining methodological issues should be addressed.

Response: Thank you for your valuable comment.

2. Tables 2 and 3. Displaying p-values < .05 in red is inappropriate. There is nothing sacred about the conventional cut-off .05, and its common use to represent “significance” is bad scientific practice. The interpretation of strength of evidence for refuting a null hypothesis

provided by a p-value depends on sample size, parameter estimates, and model assumptions. For example, the estimated HR for tumor size in Table 1 of 1.014, with 95% CI 1.000 – 1.028, is trivially different from 1, despite the fact that the p-value 0.048 is nominally significant. In fact, the p-value = .048 gives S-value = 4 which, as refutational evidence for the null hypothesis that this HR = 1, is as surprising as flipping a coin that one believes is fair 4 times and observing 4 heads.

Response: Thank you for your valuable comment. We agreed with your opinions. We have removed the red marks in **Table 2** and **Table 3**.

3. Reporting p-values computed in two different ways in Table 3 is redundant and a distraction. In any case, the distribution theory underlying the chi-square tests is asymptotic, which renders the numerical p-values unreliable given the small event counts.

Response: Thank you for your valuable comment. We have removed the p-values for chi-square test in **Table 3**.

4. In general, if multiple p-values are reported in an analysis, they should be adjusted upward to correct for multiple testing to control the overall false positive rate. In each table that gives multiple p-values, these should be replaced by upwardly adjusted values. There are many methods to adjust for multiple testing (Bonferroni, Holm, etc.). See, for example, Chen et al. J Thoracic Disease, Vol 9, No 6, 2017. The need to upwardly adjust p-values for multiple tests is due to the fact that the overall false positive rate is inflated. For example, if 20 tests are performed, the null probability that at least one of the 20 tests will have p-value < .05

equals .64. Consequently, when multiple p-values are presented, reporting unadjusted values is not correct and is misleading.

Response: Thank you for your valuable comment. According to your comments, we used the Benjamini–Hochberg method (BH method) to control the false discovery rate (FDR)¹⁻². The detailed operation is as follows.

The present study aimed to compare the long-term outcomes of robotic distal gastrectomy (RDG) and laparoscopic distal gastrectomy (LDG). The prespecified endpoints in the present study included 3-year disease-free survival, 3-year overall survival and recurrence patterns. Therefore, the three p values were adjusted by BH method. In addition, we also adjusted the p values in the subgroup analysis of the 3 endpoints independently. The above description was supplemented in the **Methods section** with red mark.

The adjusted p values were displayed in all survival curves. In addition, the p values of subgroup analysis in Fig. 4 and competing risk analysis in Table 3 were also adjusted.

1. Chen SY, Feng Z, Yi X. A general introduction to adjustment for multiple comparisons. *J Thorac Dis* 2017; 9(6):1725-1729.

2. Benjamini Y, Hochberg Y. Controlling the false discovery rate: a practical and powerful approach to multiple testing. *Journal of the Royal statistical society: series B (Methodological)* 1995; 57(1):289-300.

5. The table of subgroup-treatment interaction HR estimates giving a forest plot of 17 different confidence intervals, also suffers from the problem of multiple testing. Only adjusted p-values should be reported there.

Response: Thank you for your valuable comment. According to your comments, we only

reported the adjusted p-values in Fig. 4.

6. It still appears to me that the randomization was restricted to achieve balance, as shown by the figure in which 300 patients were randomized with 150 to LADG and 150 to RADG. This sample size outcome is extremely unlikely if a simple fair randomization was performed independently for each patient at enrollment. This is not a major point, but I find the sample sizes puzzling.

Response: Thank you for your valuable comment. We are sorry that we did not understand your comments in our last response. As you commented, the randomization was restricted to achieve balance. The specific operation of the randomization is as follows.

Eligible patients were randomly allocated in a 1:1 ratio to either LDG or RDG group. The data manager (R.H.T.), who was not involved in the eligibility assessment and recruitment of patients, conducted the randomization using a randomly ordered treatment identifier list

generated by a permuted block design with a block size of six using SAS software (version 9.2, SAS Institute Inc.). This implies that every six patients form a randomized block, with the treatment order within each block being random, and both treatment LAG and treatment RDG accounting for half of each block. Subsequently, these blocks are concatenated to form a list, and the treatment assignment for each patient is done sequentially according to this list.

The above description was supplemented in the **Methods section** with red mark.

Reviewer #3 (Remarks to the Author):

The authors appropriately responded to the reviewer's comments.

Response: Thank you for your valuable comment.

In conclusion, we have checked the manuscript and revised it according to all the comments. We submit here the revised manuscript as well as a list of changes. If you have any question about this manuscript, please don't hesitate to let me know.

Sincerely yours,

Prof. Chang-Ming Huang

Department of Gastric Surgery, Fujian Medical University Union Hospital,

E-mail: hcmlr2002@163.com

Reviewers' Comments:

Reviewer #1:

Remarks to the Author:

Concerns have been addressed.

Reviewer #2:

Remarks to the Author:

The authors' responses to my earlier comments are acceptable.